

# Staying on top of SMEFT-likelihood analyses

**Nina Elmer, Maeve Madigan, Tilman Plehn and Nikita Schmal**

Institut für Theoretische Physik, Universität Heidelberg, Germany

## Abstract

We present a new global SMEFT analysis of LHC data in the top sector. After updating our set of measurements, we show how public ATLAS likelihoods can be incorporated into an external global analysis and how our analysis benefits from the additional information. We find that, unlike for the Higgs and electroweak sector, the SMEFT analysis of the top sector is mostly limited by the theory uncertainties. Finally, we present the first global SFITTER analysis combining the top and electroweak-Higgs sectors.

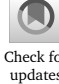
doi:10.21468/SciPostPhys.18.3.108

# 1 Introduction

Over the last decade, LHC physics has seen a paradigm shift, from testing models for physics beyond the Standard Model (BSM) to precision measurements and a complete understanding of LHC physics in terms of fundamental quantum field theory. This change represents impressive progress in experiment and theory in a holistic approach to the huge LHC dataset. On the theory side, a driving force is the development of an effective field theory version of the Standard Model (SMEFT), which allows us to ask and answer the question: *Does the LHC data agree with the Standard Model altogether?*

SMEFT is a perturbative quantum field theory that respects the gauge symmetries and covers all sectors of the Standard Model. It is renormalizable and allows for QCD and electroweak precision predictions. It is built on the idea that BSM particles affecting LHC measurements might be too heavy to be produced on-shell. Assuming that the Higgs and Goldstone fields form the usual doublet, SMEFT embeds the Standard Model in an effective field theory (EFT). The SMEFT idea [1–4] was developed for a gauge-invariant description of anomalous gauge interactions at LEP [5,6]. Its big success is the unified global analysis of the Higgs and electroweak sector, including electroweak precision data [7–9], a major part of the legacy of the LHC Run 2 [9–13].Obviously, the same approach [14–16] can be used to systematically test the top quark sector [17–27]. This sector is especially interesting because it can be combined with the bottom sector [28–33] and a much broader set of precision measurements [34], eventually testing the impact of flavor symmetries. Recently, several groups have provided combined SMEFT analyses of the electroweak and top sectors [35,36], SMEFT analyses combined with parton density extraction [37–40], and even SMEFT analyses with lighter new particles [41–43].

The systematic search for BSM effects in the top sector has some unique aspects. Experimentally, precision measurements at the LHC go far beyond simple kinematic distributions of top pair production. Associated top pair production with gauge bosons, single top production, and top decay kinematics are also probed with increasing precision. This effort is beautifully matched by precision predictions [44]. Theoretically, the top sector is still closely related to the hierarchy problem or the dynamic origin of the Higgs VEV, a problem which should be understood at the LHC [45]. Phenomenologically, the ATLAS and CMS top groups are providing experimental results in a way that can be implemented in an external global analysis easily and optimally. This includes unfolded rate measurements, unfolded kinematic distributions, and most recently published likelihoods [46–48].

The publication of experimental data in the format of public likelihoods is a major step in the way experimental results can be re-interpreted [49–54]. The HistFactory format [55], software such as `pyhf` [56,57], `Spey` [58], and simplified likelihood frameworks [59] allow for an efficient use of likelihoods. In the classic BSM sector, many likelihoods have already been made public and analyzed [60]. In the top sector, public ATLAS likelihoods [46–48] still have to be used outside the collaborations. We aim to fill this gap by using them as the basis of a global SMEFT analysis of the top sector using SFITTER.

In this paper, we update an earlier NLO-SMEFT analysis of the top sector in the SFITTER framework [21]. SFITTER is unique in the sense, that it does not rely on pre-processed experimental measurements and includes its own comprehensive uncertainty treatment [61–63]. This makes it a promising candidate to, for the first time, include public likelihoods in a global analysis and determine their impact. After a general introduction to the SFITTER methodology and our dataset in Sec. 2, we will discuss three public likelihoods in the top sector in detail in Sec. 3. We will then include these likelihoods in the first SFITTER analysis of the electroweak and top sectors in Sec. 4. While the physics behind combining these two sectors is largely understood [35,36], in our global SFITTER analysis, we will focus on the impact of

theory uncertainties. In addition, we will probe the impact of a profile likelihood vs Bayesian marginalization when extracting limits on single Wilson coefficients, where we saw significant effects on the Higgs and electroweak sector [64]. Finally, we will provide a short comparison between the Markov chains used in this analysis and the Monte Carlo experiment method used in earlier SFITTER analyses in the Appendix.

## 2 Setup

### 2.1 SMEFT Lagrangian

By fundamental theory arguments, the SMEFT Lagrangian is the appropriate interpretation framework to interpret LHC searches for effects of particles which are too heavy to be produced on-shell [4]. While in the Higgs sector one can argue about the proper way to implement electroweak symmetry-breaking and the doublet nature of the Higgs and Goldstone fields, the SMEFT description of the top sector is fixed. An open question is how to combine it with the light-flavor sector and its range of potential global symmetries. This renders the impact of flavor measurements on the top sector somewhat unclear, so we will not exploit this link and instead refer to dedicated analyses [28–32].

The goal of our analysis is to probe effective higher-dimensional interactions in the top sector using an increasing set of LHC measurements [21,65]. Because at dimension six the set of allowed operators already exceeds the power of the available measurements, we truncate the effective Lagrangian

$$\mathscr{L}_{\text{eff}} = \sum_j \left( \frac{C_j}{\Lambda^2} {}^{\ddagger}\mathcal{O}_j + \text{h.c.} \right) + \sum_k \frac{C_k}{\Lambda^2} \mathcal{O}_k . \tag{1}$$

This means the sum runs over all operators at mass dimension six, involving top quarks. Non-hermitian operators are denoted as ${}^{\ddagger}\mathcal{O}$. We neglect the Weinberg operator at dimension 5, as well as all operators of mass dimension seven and higher in the EFT expansion, assuming that their $\Lambda$-suppression translates into a suppression of their effects on LHC observables. This assumption is formally well-motivated but given the rather modest scale separation between the LHC and the accessible $\Lambda$-values, it has to be checked for a given dataset and a given UV-completion matched to the SMEFT Lagrangian [66–70].

Because the underlying symmetry structure is an input to an EFT construction, and we are hesitant to leave the test of fundamental symmetries to a numerically tricky and hardly conclusive global analysis [71], we ignore CP-violating operators. Finally, the fact that the top-sector measurements included in our analysis are blind to the light-quark flavor we assume separate $U(2)$ symmetries in the first and second generation [72,73],

$$
\begin{aligned}
q_i &= (u_L^i, d_L^i), & u_i &= u_R^i, & d_i &= d_R^i, & \text{for} \quad i = 1, 2, \\
Q &= (t_L, b_L), & t &= t_R, & b &= b_R.
\end{aligned}
\tag{2}
$$

All quark masses except for the top mass are assumed to be zero.

Our assumptions leave us with 22 independent operators in the top sector. Eight operators come with a chiral $LL$ or $RR$ structure of interacting fermion currents

$$
\begin{aligned}
\mathcal{O}_{Qq}^{1,8} &= (\bar{Q}\gamma_\mu T^A Q)(\bar{q}_i \gamma^\mu T^A q_i), & \mathcal{O}_{Qq}^{1,1} &= (\bar{Q}\gamma_\mu Q)(\bar{q}_i \gamma^\mu q_i), \\
\mathcal{O}_{Qq}^{3,8} &= (\bar{Q}\gamma_\mu T^A \tau^I Q)(\bar{q}_i \gamma^\mu T^A \tau^I q_i), & \mathcal{O}_{Qq}^{3,1} &= (\bar{Q}\gamma_\mu \tau^I Q)(\bar{q}_i \gamma^\mu \tau^I q_i), \\
\mathcal{O}_{tu}^{8} &= (\bar{t}\gamma_\mu T^A t)(\bar{u}_i \gamma^\mu T^A u_i), & \mathcal{O}_{tu}^{1} &= (\bar{t}\gamma_\mu t)(\bar{u}_i \gamma^\mu u_i), \\
\mathcal{O}_{td}^{8} &= (\bar{t}\gamma^\mu T^A t)(\bar{d}_i \gamma_\mu T^A d_i), & \mathcal{O}_{td}^{1} &= (\bar{t}\gamma^\mu t)(\bar{d}_i \gamma_\mu d_i).
\end{aligned}
\tag{3}
$$

Table 1: Wilson coefficients and their contributions to top observables via SM-interference ($\Lambda^{-2}$) and via dimension-6 squared terms only ($\Lambda^{-4}$). A square bracket indicates that the Wilson coefficient contributes to the interference at NLO in QCD. Table adapted from Ref. [64].

| Wilson coeff | | $t\bar{t}$ | single $t$ | $tW$ | $tZ$ | $t$-decay | $t\bar{t}Z$ | $t\bar{t}W$ |
|---|---|---|---|---|---|---|---|---|
| $C_{Qq}^{1,8}$ | | $\Lambda^{-2}$ | – | – | – | – | $\Lambda^{-2}$ | $\Lambda^{-2}$ |
| $C_{Qq}^{3,8}$ | | $\Lambda^{-2}$ | $\Lambda^{-4}[\Lambda^{-2}]$ | – | $\Lambda^{-4}[\Lambda^{-2}]$ | $\Lambda^{-4}[\Lambda^{-2}]$ | $\Lambda^{-2}$ | $\Lambda^{-2}$ |
| $C_{tu}^8, C_{td}^8$ | Eq.(3) | $\Lambda^{-2}$ | – | – | – | – | $\Lambda^{-2}$ | – |
| $C_{Qq}^{1,1}$ | | $\Lambda^{-4}[\Lambda^{-2}]$ | – | – | – | – | $\Lambda^{-4}[\Lambda^{-2}]$ | $\Lambda^{-4}[\Lambda^{-2}]$ |
| $C_{Qq}^{3,1}$ | | $\Lambda^{-4}[\Lambda^{-2}]$ | $\Lambda^{-2}$ | – | $\Lambda^{-2}$ | $\Lambda^{-2}$ | $\Lambda^{-4}[\Lambda^{-2}]$ | $\Lambda^{-4}[\Lambda^{-2}]$ |
| $C_{tu}^1, C_{td}^1$ | | $\Lambda^{-4}[\Lambda^{-2}]$ | – | – | – | – | $\Lambda^{-4}[\Lambda^{-2}]$ | – |
| $C_{Qu}^8, C_{Qd}^8$ | | $\Lambda^{-2}$ | – | – | – | – | $\Lambda^{-2}$ | – |
| $C_{tq}^8$ | Eq.(4) | $\Lambda^{-2}$ | – | – | – | – | $\Lambda^{-2}$ | $\Lambda^{-2}$ |
| $C_{Qu}^1, C_{Qd}^1$ | | $\Lambda^{-4}[\Lambda^{-2}]$ | – | – | – | – | $\Lambda^{-4}[\Lambda^{-2}]$ | – |
| $C_{tq}^1$ | | $\Lambda^{-4}[\Lambda^{-2}]$ | – | – | – | – | $\Lambda^{-4}[\Lambda^{-2}]$ | $\Lambda^{-4}[\Lambda^{-2}]$ |
| $C_{\phi Q}^-$ | | – | – | – | $\Lambda^{-2}$ | – | $\Lambda^{-2}$ | – |
| $C_{\phi Q}^3$ | | – | $\Lambda^{-2}$ | $\Lambda^{-2}$ | $\Lambda^{-2}$ | $\Lambda^{-2}$ | $\Lambda^{-2}$ | – |
| $C_{\phi t}$ | | – | – | – | $\Lambda^{-2}$ | – | $\Lambda^{-2}$ | – |
| $C_{\phi tb}$ | Eq.(5) | – | $\Lambda^{-4}$ | $\Lambda^{-4}$ | $\Lambda^{-4}$ | $\Lambda^{-4}$ | – | – |
| $C_{tZ}$ | | – | – | – | $\Lambda^{-2}$ | – | $\Lambda^{-2}$ | – |
| $C_{tW}$ | | – | $\Lambda^{-2}$ | $\Lambda^{-2}$ | $\Lambda^{-2}$ | $\Lambda^{-2}$ | – | – |
| $C_{bW}$ | | – | $\Lambda^{-4}$ | $\Lambda^{-4}$ | $\Lambda^{-4}$ | $\Lambda^{-4}$ | – | – |
| $C_{tG}$ | | $\Lambda^{-2}$ | $[\Lambda^{-2}]$ | $\Lambda^{-2}$ | – | $[\Lambda^{-2}]$ | $\Lambda^{-2}$ | $\Lambda^{-2}$ |

Six operators show a *LR* or *RL* chirality in the current-current interaction,

$$\mathcal{O}_{Qu}^8 = (\bar{Q}\gamma^\mu T^A Q)(\bar{u}_i \gamma_\mu T^A u_i), \qquad \mathcal{O}_{Qu}^1 = (\bar{Q}\gamma^\mu Q)(\bar{u}_i \gamma_\mu u_i),$$
$$\mathcal{O}_{Qd}^8 = (\bar{Q}\gamma^\mu T^A Q)(\bar{d}_i \gamma_\mu T^A d_i), \qquad \mathcal{O}_{Qd}^1 = (\bar{Q}\gamma^\mu Q)(\bar{d}_i \gamma_\mu d_i), \qquad (4)$$
$$\mathcal{O}_{tq}^8 = (\bar{q}_i \gamma^\mu T^A q_i)(\bar{t}\gamma_\mu T^A t), \qquad \mathcal{O}_{tq}^1 = (\bar{q}_i \gamma^\mu q_i)(\bar{t}\gamma_\mu t).$$

Finally, there are eight operators, which couple two heavy quarks to the gauge bosons [74],

$$\mathcal{O}_{\phi Q}^1 = (\phi^\dagger i \overleftrightarrow{D_\mu} \phi)(\bar{Q}\gamma^\mu Q), \qquad {}^\ddagger\mathcal{O}_{tB} = (\bar{Q}\sigma^{\mu\nu} t)\,\widetilde{\phi}\, B_{\mu\nu},$$
$$\mathcal{O}_{\phi Q}^3 = (\phi^\dagger i \overleftrightarrow{D_\mu^I} \phi)(\bar{Q}\gamma^\mu \tau^I Q), \qquad {}^\ddagger\mathcal{O}_{tW} = (\bar{Q}\sigma^{\mu\nu} t)\,\tau^I \widetilde{\phi}\, W_{\mu\nu}^I,$$
$$\mathcal{O}_{\phi t} = (\phi^\dagger i \overleftrightarrow{D_\mu} \phi)(\bar{t}\gamma^\mu t), \qquad {}^\ddagger\mathcal{O}_{bW} = (\bar{Q}\sigma^{\mu\nu} b)\,\tau^I \phi\, W_{\mu\nu}^I, \qquad (5)$$
$$^\ddagger\mathcal{O}_{\phi tb} = (\widetilde{\phi}^\dagger i D_\mu \phi)(\bar{t}\gamma^\mu b), \qquad {}^\ddagger\mathcal{O}_{tG} = (\bar{Q}\sigma^{\mu\nu} T^A t)\,\widetilde{\phi}\, G_{\mu\nu}^A.$$

The relation of these operators with the Warsaw basis [75] is worked out in the appendix of Ref. [21].

The interactions with the physical states are given by the gauge structure of the electroweak SM, so we use the combinations

$$C_{\phi Q}^\pm = C_{\phi Q}^1 \pm C_{\phi Q}^3, \qquad \text{and} \qquad C_{tZ} = c_w C_{tW} - s_w C_{tB}. \qquad (6)$$

This way, $C_{\phi Q}^-$ and $C_{tZ}$ describe a $t\bar{t}Z$ interaction, $C_{tW}$ a $tbW$ interaction, and $C_{\phi Q}^3$ both $tbW$ and $b\bar{b}Z$ interactions. The effect of our operators on the different LHC observables are summarized in Tab. 1. Here the main question is which operators modify the LHC rate and

kinematic predictions through interference with the SM-matrix element which only contributes at dimension-6 squared order.

Further operators, which in principle affect top observables at tree level or at higher perturbative orders are strongly constrained by other observables and not included in this analysis. For instance, the ubiquitous triple-gluon coupling is strongly constrained by multi-jet production or black hole searches [76]. Operators with four heavy quarks are starting to be constrained by LHC measurements, but these modest constraints are not expected to feed back into the standard top observables: weak correlations between these operators and the remainder of the top sector of the SMEFT were observed in Ref. [38]. The effect of normalization group evolution on the Wilson coefficients of the SMEFT is also neglected here, but would be an interesting question [77].

## 2.2 Data, predictions, and uncertainties

The technical goal of our study is to integrate, for the first time, published experimental statistical models into an analysis of the SMEFT. For the purpose of this study, we analyze three measurements for which likelihoods are available in the `HistFactory` [55] format on HEPData: an ATLAS measurement of the total inclusive $t\bar{t}$ cross section [46], an ATLAS measurement

Table 2: Top pair observables included in our global analysis. 'New' is defined relative to the previous SFITTER analysis [21]. 'Likelihood' indicates a dataset for which a public likelihood is available — further details of these datasets are provided in Sec. 3.

| Experiment | | Energy [TeV] | $\mathcal{L}$ [fb$^{-1}$] | Channel | Observable | # Bins | New | Likelihood | QCD k-factor |
|---|---|---|---|---|---|---|---|---|---|
| CMS | [78] | 8 | 19.7 | $e\mu$ | $\sigma_{t\bar{t}}$ | | | | [79] |
| ATLAS | [80] | 8 | 20.2 | $lj$ | $\sigma_{t\bar{t}}$ | | | | [79] |
| CMS | [81] | 13 | 137 | $lj$ | $\sigma_{t\bar{t}}$ | | ✓ | | [79] |
| CMS | [82] | 13 | 35.9 | $ll$ | $\sigma_{t\bar{t}}$ | | | | [79] |
| ATLAS | [83] | 13 | 36.1 | $ll$ | $\sigma_{t\bar{t}}$ | | ✓ | | [79] |
| ATLAS | [84] | 13 | 36.1 | $aj$ | $\sigma_{t\bar{t}}$ | | ✓ | | [79] |
| ATLAS | [46] | 13 | 139 | $lj$ | $\sigma_{t\bar{t}}$ | | ✓ | ✓ | [79] |
| CMS | [85] | 13.6 | 1.21 | $ll, lj$ | $\sigma_{t\bar{t}}$ | | ✓ | | [85] |
| CMS | [86] | 8 | 19.7 | $lj$ | $\frac{1}{\sigma}\frac{d\sigma}{dp_T^t}$ | 7 | | | [87–89] |
| CMS | [86] | 8 | 19.7 | $ll$ | $\frac{1}{\sigma}\frac{d\sigma}{dp_T^t}$ | 5 | | | [87–89] |
| ATLAS | [90] | 8 | 20.3 | $lj$ | $\frac{1}{\sigma}\frac{d\sigma}{dm_{t\bar{t}}}$ | 7 | | | [87–89] |
| CMS | [81] | 13 | 137 | $lj$ | $\frac{1}{\sigma}\frac{d\sigma}{dm_{t\bar{t}}}$ | 15 | ✓ | | [44] |
| CMS | [91] | 13 | 35.9 | $ll$ | $\frac{1}{\sigma}\frac{d\sigma}{d\Delta y_{t\bar{t}}}$ | 8 | | | [87–89] |
| ATLAS | [92] | 13 | 36 | $lj$ | $\frac{1}{\sigma}\frac{d\sigma}{dm_{t\bar{t}}}$ | 9 | ✓ | | [44] |
| ATLAS | [93] | 13 | 139 | $aj$, high-$p_T$ | $\frac{1}{\sigma}\frac{d\sigma}{dm_{t\bar{t}}}$ | 13 | ✓ | | |
| CMS | [94] | 8 | 19.7 | $lj$ | $A_C$ | | | | [95] |
| CMS | [96] | 8 | 19.5 | $ll$ | $A_C$ | | | | [95] |
| ATLAS | [97] | 8 | 20.3 | $lj$ | $A_C$ | | | | [95] |
| ATLAS | [98] | 8 | 20.3 | $ll$ | $A_C$ | | | | [95] |
| CMS | [99] | 13 | 138 | $lj$ | $A_C$ | | ✓ | | [95] |
| ATLAS | [100] | 13 | 139 | $lj$ | $A_C$ | | ✓ | | [95] |
| ATLAS | [47] | 13 | 139 | | $\sigma_{t\bar{t}Z}$ | | ✓ | ✓ | [101] |
| CMS | [102] | 13 | 77.5 | | $\sigma_{t\bar{t}Z}$ | | | | [101] |
| CMS | [103] | 13 | 35.9 | | $\sigma_{t\bar{t}W}$ | | | | [101] |
| ATLAS | [104] | 13 | 36.1 | | $\sigma_{t\bar{t}W}$ | | ✓ | | [101] |
| CMS | [105] | 8 | 19.7 | | $\sigma_{t\bar{t}\gamma}$ | | ✓ | | |
| ATLAS | [106] | 8 | 20.2 | | $\sigma_{t\bar{t}\gamma}$ | | ✓ | | |

of the total inclusive $t\bar{t}Z$ cross section [47] and an ATLAS measurement of the total inclusive single-top cross section in the $s$-channel [48]. The implementation of these likelihoods into the SFITTER framework will be discussed in more detail in Section 3. To obtain a realistic assessment of the effect of these likelihoods on the SMEFT, we incorporate them into a global analysis.

With this in mind, our analysis in the top sector will consider all measurements listed in Tables 2 and 3. The analysis is an update to a previous global top analysis performed by SFITTER, in Ref. [21]. We highlight in Tables 2 and 3 the measurements that are new relative to those included in Ref. [21], as well as those for which a public likelihood is available. Where possible, we make use of measurements encompassing the full Run II LHC luminosity and choose measurements in the boosted regime in which sensitivity to energy-growing SMEFT operators is maximized; see, for example, the top pair production invariant mass distribution of Ref. [93]. The dataset consists of a total of 122 data points spanning the $t\bar{t}$, $t\bar{t}+X(Z,W,\gamma)$ and single top ($s$, $t$-channel, $tW$ and $tZ$) sectors, including measurements of top-pair production charge asymmetries $A_C$ and $W$ boson polarization in top decays ($F_0, F_L$).

A key ingredient to all global analyses are precision predictions from perturbative quantum field theory. Most observables considered in this analysis are unfolded to parton level, assuming stable top quarks. This allows us to use fixed-order calculations to determine the SM predictions at NLO in QCD using `MadGraph5 aMC@NLO` [131,132] and `NNPDF 4.0` [133] interfaced with `LHAPDF` [134]. Alongside the observables listed in Tables 2 and 3, we note whether the SM predictions for these observables are approximated at NNLO in QCD using a

Table 3: Single top and top decay observables included in our global analysis. 'New' is defined relative to the previous SFITTER analysis [21]. 'Likelihood' indicates a dataset for which a public likelihood is available — further details of these datasets are provided in Sec. 3.

| Exp. | $\sqrt{s}$ [TeV] | $\mathcal{L}$ [fb$^{-1}$] | Channel | Observable | # Bins | New | Likelihood | QCD k-factor |
|---|---|---|---|---|---|---|---|---|
| ATLAS [107] | 7 | 4.59 | $t$-ch | $\sigma_{tq+\bar{t}q}$ | | | | |
| CMS [108] | 7 | 1.17 ($e$), 1.56 ($\mu$) | $t$-ch | $\sigma_{tq+\bar{t}q}$ | | | | |
| ATLAS [109] | 8 | 20.2 | $t$-ch | $\sigma_{tq}, \sigma_{\bar{t}q}$ | | | | |
| CMS [110] | 8 | 19.7 | $t$-ch | $\sigma_{tq}, \sigma_{\bar{t}q}$ | | | | |
| ATLAS [111] | 13 | 3.2 | $t$-ch | $\sigma_{tq}, \sigma_{\bar{t}q}$ | | | | [112] |
| CMS [113] | 13 | 2.2 | $t$-ch | $\sigma_{tq}, \sigma_{\bar{t}q}$ | | | | [112] |
| CMS [114] | 13 | 35.9 | $t$-ch | $\frac{1}{\sigma}\frac{d\sigma}{d|p_{T,t}|}$ | 5 | ✓ | | |
| CMS [115] | 7 | 5.1 | $s$-ch | $\sigma_{t\bar{b}+\bar{t}b}$ | | | | |
| CMS [115] | 8 | 19.7 | $s$-ch | $\sigma_{t\bar{b}+\bar{t}b}$ | | | | |
| ATLAS [116] | 8 | 20.3 | $s$-ch | $\sigma_{t\bar{b}+\bar{t}b}$ | | | | |
| ATLAS [48] | 13 | 139 | $s$-ch | $\sigma_{t\bar{b}+\bar{t}b}$ | | ✓ | ✓ | |
| ATLAS [117] | 7 | 2.05 | $tW$ (2$l$) | $\sigma_{tW+\bar{t}W}$ | | | | |
| CMS [118] | 7 | 4.9 | $tW$ (2$l$) | $\sigma_{tW+\bar{t}W}$ | | | | |
| ATLAS [119] | 8 | 20.3 | $tW$ (2$l$) | $\sigma_{tW+\bar{t}W}$ | | | | |
| ATLAS [120] | 8 | 20.2 | $tW$ (1$l$) | $\sigma_{tW+\bar{t}W}$ | | ✓ | | |
| CMS [121] | 8 | 12.2 | $tW$ (2$l$) | $\sigma_{tW+\bar{t}W}$ | | | | |
| ATLAS [122] | 13 | 3.2 | $tW$ (1$l$) | $\sigma_{tW+\bar{t}W}$ | | | | |
| CMS [123] | 13 | 35.9 | $tW$ ($e\mu j$) | $\sigma_{tW+\bar{t}W}$ | | | | |
| CMS [124] | 13 | 36 | $tW$ (2$l$) | $\sigma_{tW+\bar{t}W}$ | | ✓ | | |
| ATLAS [125] | 13 | 36.1 | $tZ$ | $\sigma_{tZq}$ | | | | |
| ATLAS [126] | 7 | 1.04 | | $F_0, F_L$ | | | | |
| CMS [127] | 7 | 5 | | $F_0, F_L$ | | | | |
| ATLAS [128] | 8 | 20.2 | | $F_0, F_L$ | | | | |
| CMS [129] | 8 | 19.8 | | $F_0, F_L$ | | | | |
| ATLAS [130] | 13 | 139 | | $F_0, F_L$ | | ✓ | | |

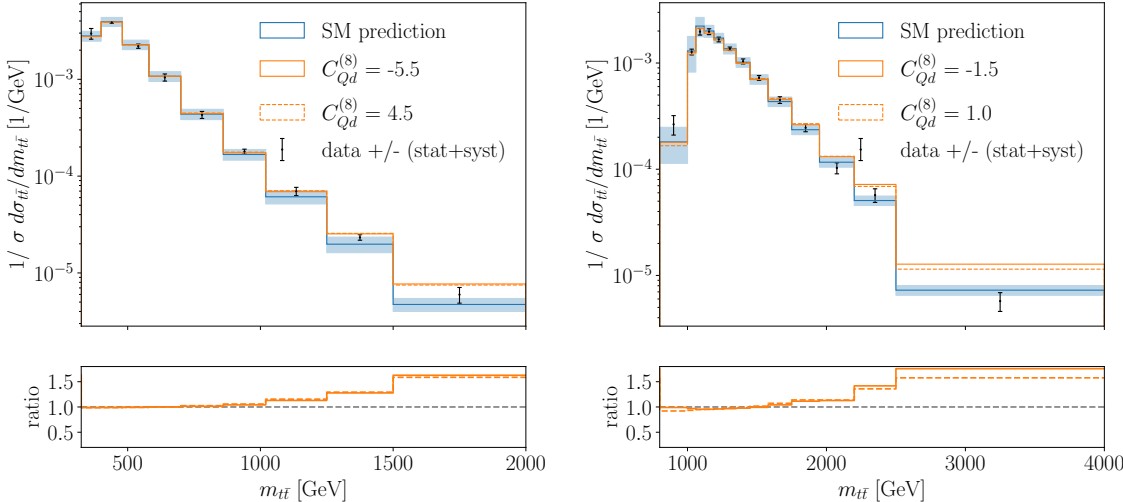

Figure 1: Left: impact of $\mathcal{O}_{Qd}^{(8)}$ on the unfolded ATLAS $m_{t\bar{t}}$ distribution in the lepton+jets channel [92]. Right: impact of this operator on the unfolded ATLAS $m_{t\bar{t}}$ distribution in the all-hadronic channel measured with boosted top quarks [93].

$K$-factor approximation and referencing the source of these QCD $K$-factors. In the case of new top quark pair production observables these QCD $K$-factors are calculated using `HighTea` [44].

Calculations of the effect of the SMEFT on all updated measurements in the top sector are performed at NLO in QCD using the FeynRules [135] model `SMEFTatNLO` [136] up to quadratic order in the EFT expansion. The exceptions are the measurements of the $t\bar{t}\gamma$ total cross sections at 8 TeV by ATLAS [106] and CMS [105], for which the SMEFT predictions at LO in QCD are taken from Ref. [38].

Theory uncertainties appear whenever we compare a measurement to a first-principle description. In principle, they cover a wide range of approximations which we make to be able to calculate, for example, an LHC cross section from a fundamental renormalized Lagrangian. For the LHC, they are dominated by the truncation of the perturbative series, in QCD and the electroweak gauge coupling. Because these perturbative series converge very slowly for LHC rates, theory predictions have become limiting factors for the interpretation of many LHC measurements in terms of actual physics. Aside from the size of the theory uncertainties, it is problematic that they do not follow any statistical pattern or model [137], and assuming a Gaussian distribution is neither justified nor conservative.

Because of their impact on global analyses of effective Lagrangians, SFITTER puts an emphasis on the proper description of these uncertainties, including their correlations between different observables. We will describe this treatment in more detail in Sec. 2.3. In the top sector, the theory uncertainties are critical for the precisely measured top pair production rates [21] and are correlated between different final states for rate measurements. We typically use the theory uncertainties reported in the respective publications, with the exception that we enforce a minimum scale uncertainty of 10% for total rates in single top production and 2% for bin-wise kinematic distributions.

**Boosted top pair production**

As part of our dataset, we highlight the reinterpretation of the ATLAS measurement of $t\bar{t}$ production in the lepton+jets channel [92] and the ATLAS measurement of $t\bar{t}$ production using boosted top quarks in the all-hadronic channel [93]. Both are differential in the top-pair

invariant mass, as shown in Fig.1. The measurement using boosted top quarks is unfolded to a fiducial parton-level phase space, defined by

$$p_{T,t_1} > 500 \text{ GeV}, \qquad \text{and} \qquad p_{T,t_2} > 350 \text{ GeV}, \qquad (7)$$

allowing for an easy comparison with fixed-order calculations. This, alongside the high-$m_{t\bar{t}}$ reach of this distribution makes it an excellent candidate for constraining the energy- growing SMEFT four-fermion operators of the top sector. We display the impact of one of these operators, $\mathcal{O}_{Qd}^8$, in Fig.1.

The theory uncertainty is shown in blue in both figures and compared to the statistical and systematic uncertainties in the experimental data. In both cases, the values of $\mathcal{C}_{Qd}^8$ chosen are those which would produce a $3\sigma$ deviation in a one-parameter analysis. We observe that, while the measurement unfolded to the full phase space is sensitive to the energy-growing effects of $\mathcal{O}_{Qd}^8$, this sensitivity is significantly enhanced by the measurement of boosted top quarks.

## 2.3 SFitter

The SFITTER framework [61–63] has been developed for global analyses of LHC measurements in the context of BSM physics and Higgs properties [7, 11, 21, 138, 139], including comprehensive studies on Higgs and electroweak properties induced by actual UV-completion of the SM [68, 140], an extrapolation for the HL-LHC [64], as well as for future electron-positron colliders [141].

The relation to full models and a proper treatment of uncertainties in precision matching is crucial for the LHC because the typical scale separation between directly probed energies and indirectly accessible energies is not very large. On the other hand, the consistency of the EFT description is not a universal property of the EFT Lagrangian, but only defined by possible on-shell propagators in the observables and relative to the UV-completion and its typical coupling strengths. Without additional information on the underlying model the Lagrangian in Eq.(1) is degenerate along $C_k \sim \Lambda^2$, which means the EFT assumption of large $\Lambda$ improves for larger postulated couplings. This is the reason why SFITTER SMEFT analyses start with the truncated dimension-6 Lagrangian at face value.

From Tab. 1, we know that some Wilson coefficients do not interfere with the SM matrix elements at leading order, so we include dimension-6 squared contributions to the LHC observables. This means we truncate the Lagrangian rather than the LHC rate prediction. We emphasize that all our assumptions are neither inherently right nor wrong, and need to be validated for a given dataset and a given UV-completion [66–68, 140]. However, our assumptions ensure that the SFITTER analysis makes optimal use of the kinematic information, especially in the tails of momentum or energy distributions.

At the heart of SFITTER is the extraction of the fully exclusive likelihood, given a rate measurement $d$ from Sec. 2.2, evaluated over the combined space of Wilson coefficients $c$ and nuisance parameters $\theta$,

$$p(d|c, \theta) = \text{Pois}(d|m(c, \theta, b)) \, \text{Pois}(b_{\text{CR}}|b\,k) \prod_i \mathcal{C}_i(\theta_i, \sigma_i). \qquad (8)$$

It incorporates the effects of the statistical, systematic, and theory uncertainties. The first Poisson distribution gives the probability to observe $d$ events given the corresponding theory prediction $m(c, \theta, b)$, which in turn depends on the predicted background count $b$. The background rate is, itself, constrained by measurements $b_{\text{CR}}$ in the control region, implemented as a scaled prediction $kb$ with a suitable factor $k$. The constraint function $\mathcal{C}$ gives the distribution of the nuisance parameter $\theta_i$, given a width measure $\sigma_i$. Depending on the source of the uncertainty, it can be chosen as follows:

- Gaussian, for systematic uncertainties related to independent measurements in other channels. Examples are other LHC rates, but also calibration. We take $\sigma_i$ from the respective experimental publications. As we will discuss in Sec. 3, public likelihoods will help here.

- Flat, for theory uncertainties which do not have a well-defined maximum and could be thought of as a range [137]. Examples are scale uncertainties for QCD predictions and PDF uncertainties. They are usually taken from the experimental publications, but we increase them whenever the standard choice appears not conservative.

The flat scale uncertainty is not parametrization invariant, as one would expect from a fixed range, but without a preferred central value we consider it conservative. Scale uncertainties are obtained by varying the renormalization and factorization scales $\mu_R$ and $\mu_F$ by a factor of 2 around their respective central value. These are process dependent and chosen to be $\mu_R = \mu_F = m_t + \frac{1}{2}m_V$ for associated $t\bar{t}$ production with $V = W, Z$. For $t\bar{t}$ production, the sum of the transverse masses of the top and anti-top is used, while for single top production they are set to the top mass $m_t$.

By ansatz, SFITTER treats all measurements $d$ as uncorrelated, constructing an individual likelihood for each measurement, as defined in Eq.(8). This assumption is justified by the individual statistical uncertainties described by the Poisson distributions in the likelihood. The full likelihood can thus be constructed as the product of these individual contributions. As a consequence, the Gaussian constraint terms describing the systematics of each measurement can be generalized to a single higher-dimensional Gaussian which allows correlations between uncertainties to be introduced. In the case of Gaussian systematics, the correlations are described by a correlation matrix [142]

$$C_{ij} = \frac{\sum_{\text{syst}} \rho_{ij} \sigma_{i,\text{syst}} \sigma_{j,\text{syst}}}{\sigma_{i,\text{exp}} \sigma_{j,\text{exp}}}, \qquad \text{with} \qquad \sigma_{i,\text{exp}}^2 = \sum_{\text{syst}} \sigma_{i,\text{syst}}^2 + \sum_{\text{Poiss}} \sigma_{i,\text{Poiss}}^2. \qquad (9)$$

Here $i, j$ run over all measurements and $\sigma_{\text{syst}}, \sigma_{\text{Poiss}}$ are the systematic and Poisson uncertainties. We then choose $\rho_{ij}$ to be either uncorrelated or (essentially) fully correlated for systematics of the same type.

Theory uncertainties are correlated for all measurements with identical predictions. They are also correlated within one measurement across all bins but not across several different measurements. This is done by averaging them, weighted such that the final standard deviation is minimized. Using the prediction only once for this weighted average, instead of each individual measurement, ensures the proper correlations of the corresponding theory uncertainties. The implementation of flat theory uncertainties allows for a shift of the prediction within their bounds without affecting the likelihood value.

To construct the exclusive likelihood, SFITTER uses cross section predictions over the entire model parameter space and extracts the quadratic behavior analytically, which guarantees sufficient precision even for small Wilson coefficients. We then use a Markov chain to evaluate Eq.(8) numerically and to encode the likelihood in the distribution of points covering the combined space of Wilson coefficients and nuisance parameters. The setup of the Markov chain depends on the structure of the model space; for BSM analyses an efficient search for local structures in the global model space is important, while for SMEFT analyses we know that the global likelihood maximum will be close to the SM-point. Adjusting the Markov chain accordingly leads to a significant speed improvement [68]. Our motivation to use a Markov chain rather than so-called toys is described in the Appendix.

Finally, to combine uncertainties by removing nuisance parameters or to reduce the space of physical Wilson coefficients, SFITTER can employ a profile likelihood or a Bayesian marginalization [62, 64]. Obviously, these two methods give different results. Only for uncorrelated

Gaussians do the profile likelihood and Bayesian marginalization lead to the common result of errors added in quadrature. For a flat likelihood, the uncorrelated profile likelihood adds the two uncertainties linearly, which happens for the scale uncertainty and the PDF uncertainty in SFITTER. The profile likelihood combination of a flat and a Gaussian uncertainty gives the well-known RFit prescription [143]. In contrast, when applying marginalization on the combination of Gaussian and flat uncertainties, the central limit theorem ensures that the final posterior will be Gaussian again.

## 3 Public likelihoods

For a standard SFITTER analysis, we extract systematic uncertainties for each measurement from the respective experimental publications. Systematics of the same type are fully correlated between measurements of the same experiment. This approach has drawbacks, for instance, we can only use the uncertainty categories reported in the experimental publications or on HEPData, and this information often needs to be extracted by hand. Public likelihoods include the full information on a large number of systematic uncertainties in a documented manner, making their implementation more accurate and efficient.

Likelihoods are published in the HistFactory format [55], similar to the SFITTER likelihood in Eq.(8). For each bin $b$ measured in a kinematic distribution of a given channel or final state, it provides

$$p(d_b|\mu, \theta) = \text{Pois}(d_b|m_b(\mu, \theta)) \prod_i \mathcal{C}_i(a_i|\theta_i), \tag{10}$$

where $d_b$ and $m_b$ are the measured and expected number of events in bin $b$. The nuisance parameters $\theta_i$ are constrained by $\mathcal{C}_i(a_i|\theta_i)$ with the auxiliary data $a_i$. The parameter of interest $\mu$ describes, for instance, a signal strength. It corresponds to the Wilson coefficient in Eq.(8).

The analysis of these likelihoods is performed using pyhf [56,57], a python module allowing for easy construction of HistFactory likelihoods and their subsequent statistical analysis. It uses data published in the JSON format to compute the predicted number of events using

$$m_b = \sum_s \left( \prod_\kappa \kappa_{sb} \right) \left( \bar{m}_{sb} + \sum_\Delta \Delta_{sb} \right), \tag{11}$$

Table 4: List of modifiers in the construction of the HistFactory likelihoods, adapted from Ref. [144]. Per-bin modifiers are denoted as $\gamma_b$, while interpolated modifiers are denoted as $\alpha$. Here $g_p$ and $f_p$ describe different interpolation strategies used to compute these from the values $\kappa_{sb,\alpha=\pm1}, \Delta_{sb,\alpha=\pm1}$ provided in the likelihood. Luminosity and scale factors affect all bins equally and are denoted as $\lambda$ and $\mu$, respectively.

| Description | Modification | Constraint $\mathcal{C}$ |
|---|---|---|
| Luminosity ('lumi') | $\kappa_{sb} = \lambda$ | $\mathcal{N}(l = \lambda_0|\lambda, \sigma_\lambda)$ |
| Normalization unc. ('normsys') | $\kappa_{sb} = g_p(\alpha|\kappa_{sb,\alpha=\pm1})$ | $\mathcal{N}(a = 0|\alpha, \sigma = 1)$ |
| Correlated Shape ('histosys') | $\Delta_{sb} = f_p(\alpha|\Delta_{sb,\alpha=\pm1})$ | $\mathcal{N}(a = 0|\alpha, \sigma = 1)$ |
| MC Stat. ('staterror') | $\kappa_{sb} = \gamma_b$ | $\prod_b \mathcal{N}(a_{\gamma_b} = 1|\gamma_b, \delta_b)$ |
| Uncorrelated Shape ('shapesys') | $\kappa_{sb} = \gamma_b$ | $\prod_b \text{Pois}(\sigma_b^{-2}|\sigma_b^{-2}\gamma_b)$ |
| Normalization ('normfactor') | $\kappa_{sb} = \mu$ | |

with the nominal expected rate $\bar{m}_{sb}$ and multiplicative ($\kappa_{sb}$) and additive ($\Delta_{sb}$) modifiers for each physics process $s$. These modifiers correspond to the nuisance parameters affecting the event rate $m_b$. The type of modifier and the constraints on its corresponding nuisance parameter depend on the type of uncertainty. The most common are given in Tab. 4. Using the public likelihoods in terms of modifiers and nominal rates $\bar{m}_{sb}$, we can reproduce the experimental results. For visualization, we use `cabinetry` [145], a python library making use of `pyhf` for statistical analyses.

Starting from the public likelihoods, we organize the full set of nuisance parameters corresponding to systematic uncertainties in a small number of categories. This allows for an easier numerical treatment at, essentially, no cost. To compute the ranges of nuisance parameters for these categories, we first use a profile likelihood to determine the central values and, in a second step, an analysis of the distribution of the nuisance parameter. In this section, we will show how we implement and test three public ATLAS likelihoods.

## 3.1 ATLAS $t\bar{t}$ likelihood

The first public likelihood we analyze covers the $t\bar{t}$ rate measurement in the leptons+jets final state [46]. It consists of three channels or signal regions, using the aplanarity, minimum lepton-jet mass and average angular distance between jets. The parameter of interest $\mu$ is the $t\bar{t}$ signal strength, with a total of 177 nuisance parameters covering the systematic uncertainties.

To test our implementation and evaluation of the public likelihood, we first reproduce some key ATLAS results in Fig. 2. We show the values for each nuisance parameter that maximizes the likelihood and the pulls,

$$\text{pull} = \frac{\hat{\theta} - \theta_0}{\Delta\theta} \, . \tag{12}$$

Here, $\hat{\theta}$ describes the maximum likelihood values and $\theta_0$ is the value before the fit, normalized to the pre-fit uncertainty $\Delta\theta$. We also show the impact of the individual nuisance parameters on the signal strength $\mu$. It is determined by repeating the fit after fixing the nuisance parameter to its maximum-likelihood value $\hat{\theta}$, shifted by its prefit (postfit) uncertainties $\pm\Delta\theta(\pm\Delta\hat{\theta})$. The left panel of Fig. 2 is taken from Ref. [46], while the right panel shows our reproduced results. Both sets show excellent agreement, with negligible differences for a few select nuisance parameters.

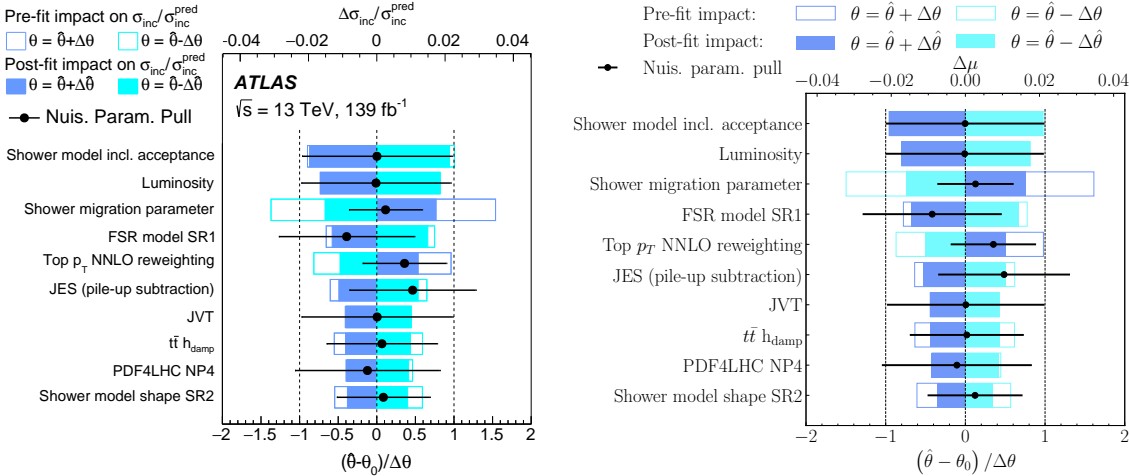

Figure 2: Impact of nuisance parameters on the $t\bar{t}$ total rate fit. We compare the ATLAS result [46] (left) and our evaluation of the public likelihood (right).

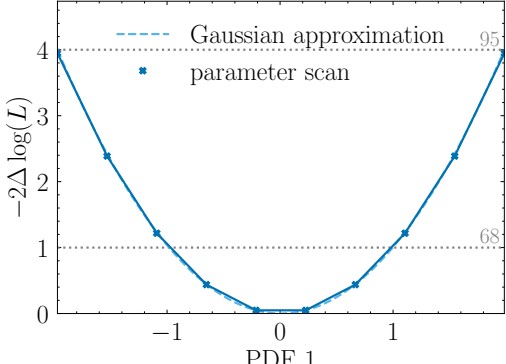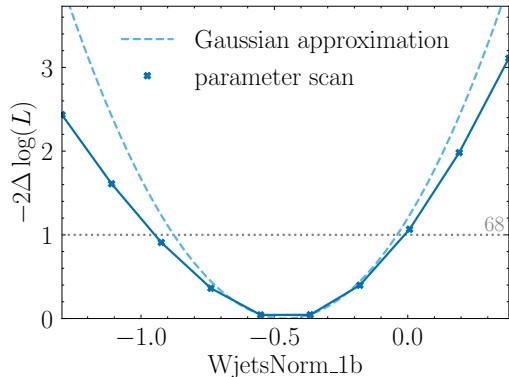

Figure 3: Dependence of the log-likelihood for the $t\bar{t}$ rate on two nuisance parameters, one describing the PDF uncertainty and one describing the $W+$ jets background normalization uncertainty, compared with the Gaussian approximation.

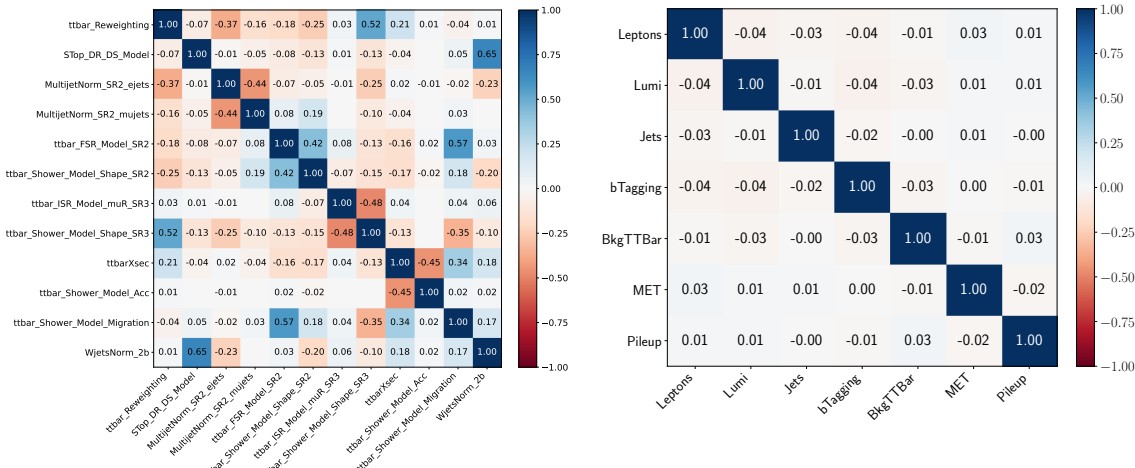

Figure 4: Left: Correlations between individual nuisance parameters affecting the $t\bar{t}$ rate with at least one correlation greater than 0.4. Right: Correlations between categories of systematic uncertainties extracted from the $t\bar{t}$ likelihood as implemented in SFITTER.

Next, we analyze the full likelihood as a function of a single nuisance parameter. This allows us to check the validity of a Gaussian likelihood, as assumed for systematic uncertainties in SFITTER. For each nuisance parameter, we generally find excellent agreement with the Gaussian assumption, as shown on the left, with only a few exceptions. Figure 3 shows two such cases, one with excellent agreement and one with poor agreement. Even the larger deviations are under control, showing good agreement with the Gaussian approximation when we translate them into one standard deviation. Our combination of nuisance parameters into categories washes out non-Gaussian shapes in these exceptions.

Finally, we test the correlations between individual nuisance parameters and between nuisance parameters assigned to the categories implemented in SFITTER. The left panel of Fig. 4 shows the correlations of all individual nuisance parameters with at least one correlation greater than 0.4. Since the public likelihoods do not provide additional metadata on all nuisance parameters, their labels do not necessarily match those used in the impact plots. We find that out of the many nuisance parameters included in the public likelihood, only very few are significantly correlated. We mainly see strong correlations, for instance, for modeling choices or jets.

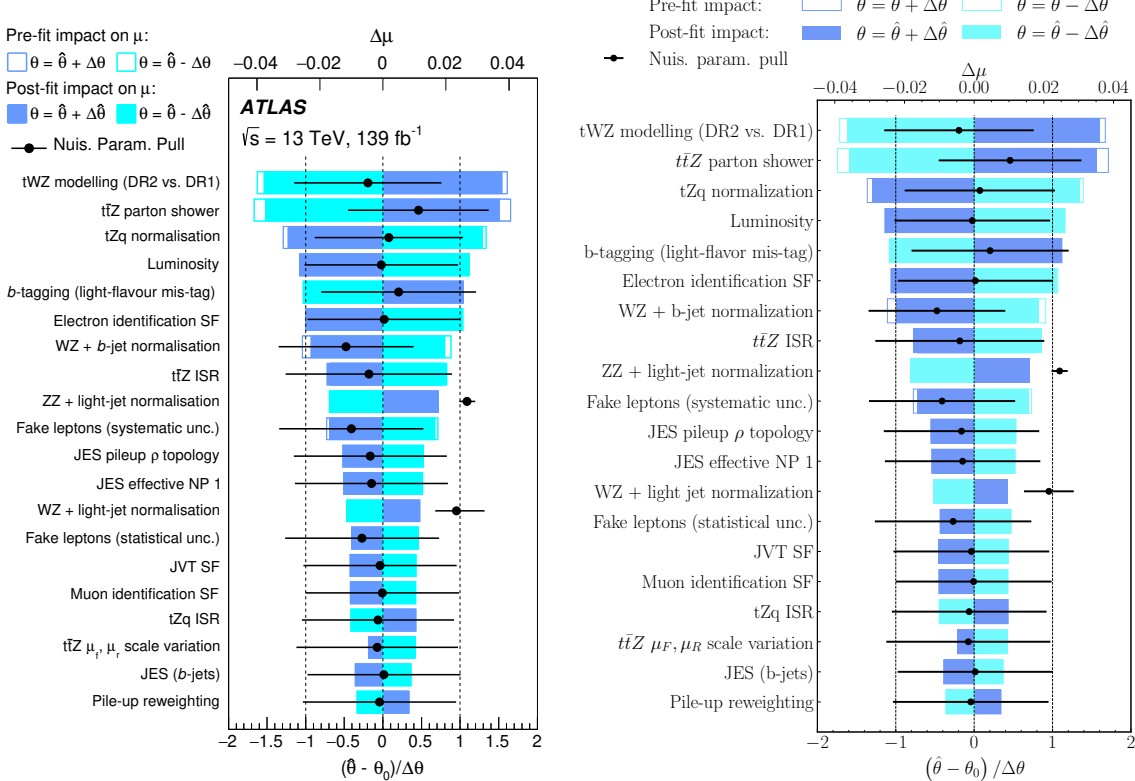

Figure 5: Impact of nuisance parameters on the $t\bar{t}Z$ total rate fit. We compare the ATLAS result [47] (left) and our evaluation of the public likelihood (right).

In the standard SFITTER approach, we group these individual nuisance parameters into uncorrelated categories and implement these categories with a single nuisance parameter each. Standard categories cover leptons, jets, tagging, and luminosity. Additional categories are process-specific, such as certain backgrounds or missing transverse energy. For all processes in our dataset, we use 21 nuisance parameters describing the systematic uncertainties assumed to be uncorrelated. Using the full likelihood, we show the correlations between these categories in the right panel of Fig. 4. The fact that the correlations between categories essentially vanish validates this SFITTER approach.

## 3.2 ATLAS $t\bar{t}Z$ likelihood

The second likelihood we implement is for the $t\bar{t}Z$ rate measurement [47]. It simultaneously fits both 3-lepton and 4-lepton signal regions and the corresponding control regions. The parameter of interest is the $t\bar{t}Z$ signal strength. A total of 230 nuisance parameters describe the systematic uncertainties. Unlike for the $t\bar{t}$ likelihood, there are no uncertainties on the shape of the signal since each signal region is described by a single bin.

Following the method described for the $t\bar{t}$ analysis, we also test the $t\bar{t}Z$ likelihood and our implementation. Figure 5 compares the impact and pulls taken from Ref. [47] with those reproduced by us. We see excellent agreement for all nuisance parameters, with, at most, very minor differences.

As before, we then show the correlations between nuisance parameters with at least one correlation greater than 0.3 in the left panel of Fig. 6. We can compare them to our SFITTER implementation on the right. We find that the correlations between individual nuisance parameters are already much smaller for this likelihood. The only strong correlations appear

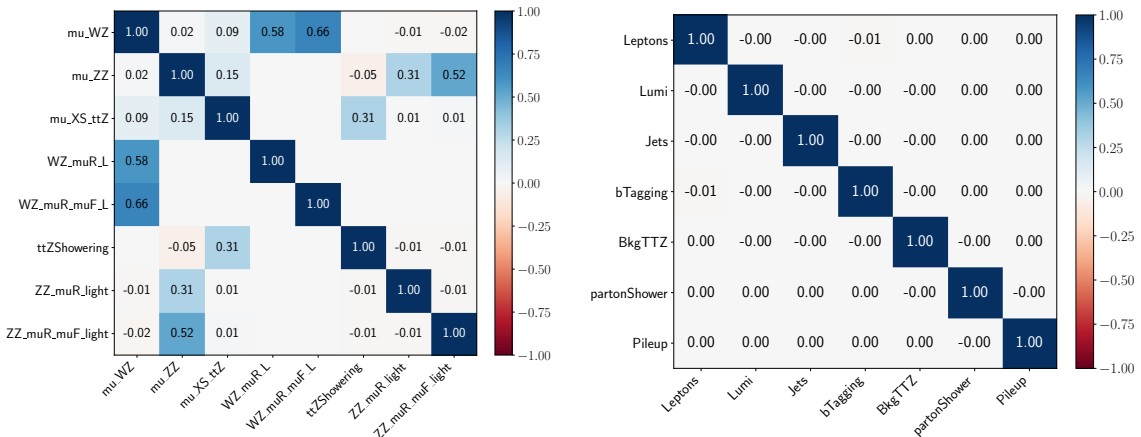

Figure 6: Left: Correlations between individual nuisance parameters affecting the $t\bar{t}Z$ rate with at least one correlation greater than 0.3. Right: Correlations between categories of systematic uncertainties extracted from the $t\bar{t}Z$ likelihood as implemented in SFITTER.

between scale uncertainties and the signal strength of the corresponding background. Consequently, the results, after combining all nuisance parameters into the SFITTER categories, display negligible correlations between categories.

## 3.3 ATLAS $s$-channel single top likelihood

The third likelihood we implement is for the signal strength of $s$-channel single top production [48]. Unlike the previous measurements, it consists of a single channel, making use of the matrix element method (MEM) to determine the probability that an event is a signal event. The discriminant defined using the MEM gives a distribution with 171 nuisance parameters affecting the rate and shape of the signal.

Once again, we validate our implementation of this likelihood in Fig. 7, showing the impact and pulls from Ref. [48] in the left panel and our reproduction on the right. We find perfect agreement, showing that regardless of the process considered, the public likelihoods allow for an easy and precise reproduction of the experimental results in more detail than most global analyses will ever need or want to use.

The correlations in the left panel of Fig. 8 show strong correlations only between select nuisance parameters. The strongest correlations appear between jet-related uncertainties and the signal strengths of the two dominant backgrounds, $t\bar{t}$ and $W$+jets. For SFITTER, these nuisance parameters are put into the background uncertainty category. These strong correlations are therefore implicitly included in this larger category, and the final implementation into SFITTER is essentially uncorrelated, as one can see in the right of Fig. 8. While one still finds a nonzero correlation between the jet and background uncertainties and between the jet and lepton uncertainties, these are all negligibly small.

## 4 Global analysis

Using, for the first time, public likelihoods in a global SMEFT analysis allows us to look at different relevant questions. From the data included in our analysis, we know that our global analysis is somewhat unlikely to uncover a fundamental and statistically significant break-down of the SM. We first look at the impact on the constraining power from new measurements, especially boosted top kinematics, relative to Ref. [21]. We then study the impact of correlated

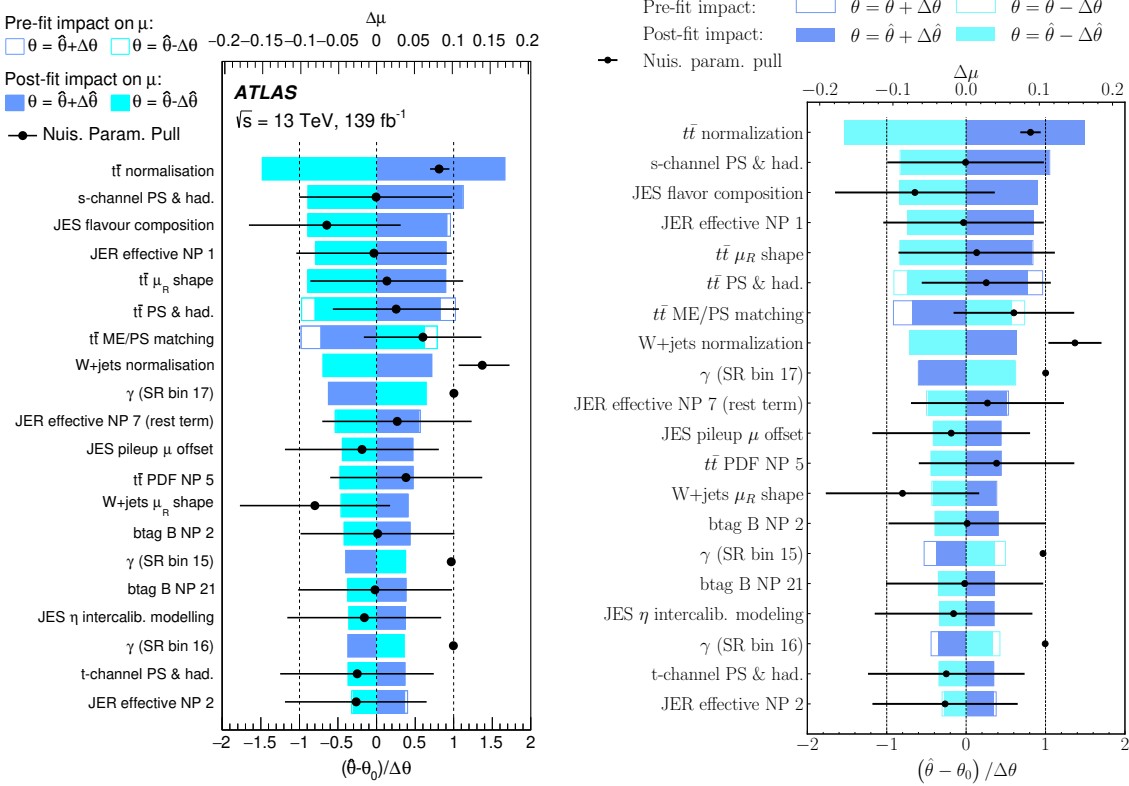

Figure 7: Impact of nuisance parameters on the *s*-channel single top rate fit. We compare the ATLAS result [48] (left) and our evaluation of the public likelihood (right).

uncertainties encoded in the public likelihoods. From a pure statistics perspective, we also check if lower-dimensional limits extracted by profiling and by marginalization differ. Finally, we provide SMEFT limits combining the updated top sector analysis with the electroweak and Higgs sector from Ref. [64].

## 4.1 Better and boosted measurements

Before we study more conceptual questions of global SMEFT analyses, we update our dataset with new measurements, as marked in Tabs. 2 and 3. In Fig. 9, we show the constraints on a selection of 2-dimensional correlations of Wilson coefficients, using all top data compared to the previous SFITTER top analysis [21]. All constraints are the result of an analysis of all 22 Wilson coefficients. To extract limits on pairs of coefficients, we use a profile likelihood. Potential differences in marginalization will be discussed in Sec. 4.3.

The left panel in Fig. 9 shows the impact on the four-fermion operators $\mathcal{O}^1_{Qu}$ and $\mathcal{O}^8_{tq}$. Both operators receive constraints from top pair production, now with a public likelihood [46], as well as new data in the boosted regime [93] and at 13.6 TeV [85]. The dominant constraining power comes from measurements of the boosted kinematics [93] and will be discussed below.

The right panel shows the improvement in constraints on $\mathcal{O}^3_{\phi Q}$ and $\mathcal{O}^{31}_{Qq}$. Single top production provides constraints on them, and we again benefit from the public likelihood [48], a new $p_{T,t}$ distribution in *t*-channel single top production [114], and new measurements of the *tW* production cross section [120, 124]. We observe an improvement in the individual constraints and their correlation. In particular, $\mathcal{O}^{31}_{Qq}$ receives some constraining power from boosted top pair production, which in turn allows single top measurements to constrain $\mathcal{O}^3_{\phi Q}$.

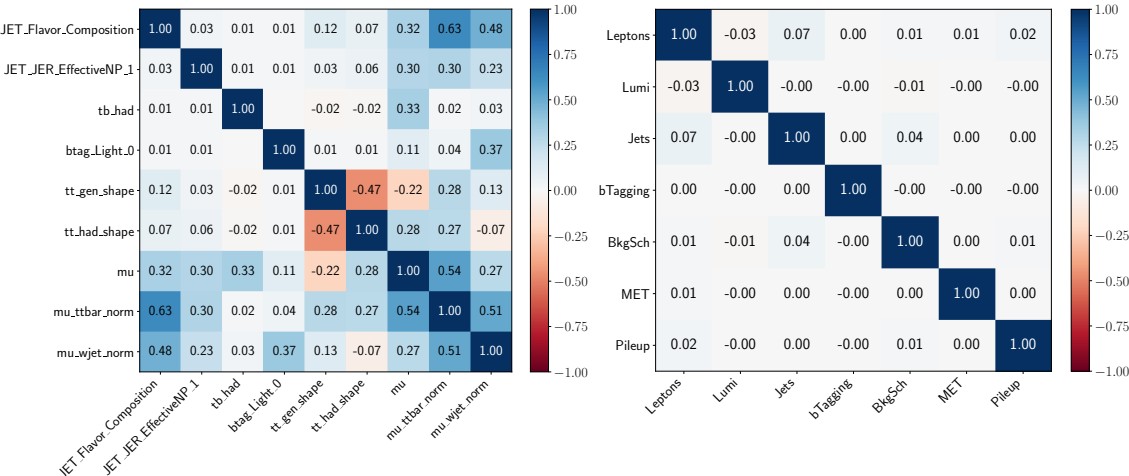

Figure 8: Left: Correlations between individual nuisance parameters affecting the $s$-channel single top rate with at least one correlation greater than 0.3. Right: Correlations between categories of systematic uncertainties extracted from the single top likelihood as implemented in SFITTER.

Finally, in the lower panel of Fig. 9 we highlight the improvement in probing $\mathcal{O}_{\phi t}$ and $\mathcal{O}_{\phi Q}^{-}$. As before, these operators are constrained by measurements of $t\bar{t}Z$ production, for which we use a public likelihood. However, in this case, we only find a small change in the correlated likelihood.

Altogether, we find that the public likelihoods do not have a significant effect on our SMEFT limits. As discussed in Sec. 3, the likelihoods available and included in our analysis all describe total cross sections, limiting their impact. On the positive side, public likelihoods allow for an accurate modeling of correlated systematics, an aspect we will discuss in Sec 4.2.

Much of the improvement we see from our new dataset is due to the boosted regime. For SMEFT analyses, such measurements are extremely helpful to constrain operators which include momentum scaling. As discussed in Sec. 2.2, we add the unfolded ATLAS measurement of boosted top pair production [93]. In Fig. 1, we already showed the impact of a single SMEFT operator $\mathcal{O}_{Qd}^8$ on the normalized $m_{t\bar{t}}$ distribution of this measurement.

Here, we study its effect on the global analysis. Figure 10 demonstrates the impact on a selection of two-operator correlations. The complete analysis including all data in Tabs. 2, Tabs. 3 is compared to the case where the measurement of boosted tops from Ref. [93] is excluded. In the first panel, we observe an increase in constraining power on $\mathcal{O}_{Qq}^{18}$, while the constraints on $\mathcal{O}_{tG}$ are stable. This follows from the fact that this operator is instead constrained by the $t\bar{t}$ total cross section. In contrast, the limits on the Wilson coefficients for energy-growing 4-fermion operators improve by a factor of two, as shown in the right-hand panel for $\mathcal{O}_{Qq}^8$ and $\mathcal{O}_{td}^8$.

## 4.2 Correlated systematics

Public likelihoods, as discussed in Sec. 3, allow us to model and study correlated systematic uncertainties across measurements by the same experiment. For measurements without public likelihoods, the approximate treatment of correlations is discussed in Sec. 2.3. For the Higgs sector, we already know that the correlations of systematic uncertainties had a highly visible impact on the SMEFT analysis [64]. In particular, they lead to a marked shift in the most likely values of Wilson coefficients while leaving the width of the limits unchanged.

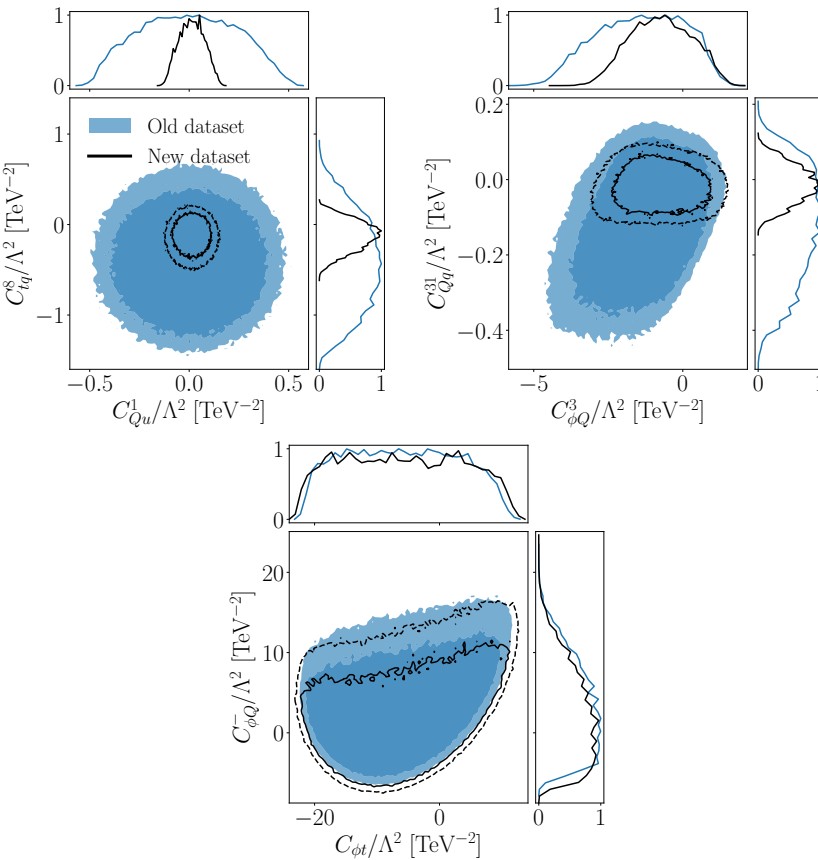

Figure 9: Profile likelihood correlations for three pairs out of the 22 Wilson coefficients, illustrating the impact of the new data listed in Tabs. 2 and 3 (black) compared to the previous analysis [21] (blue).

Here, we assess the impact of correlating systematic uncertainties on the top sector. In Fig. 11, we show two sets of constraints on a selection of Wilson coefficients. In blue, we show the constraints from a global analysis where all correlations between experimental systematics and between theory uncertainties are included. In black we show the same results, but treating all experimental systematics as uncorrelated. For all Wilson coefficients, we find good agreement, which indicates that in the top sector, the correlations of systematic uncertainties cannot be ignored but have a limited effect on the final SMEFT limits.

We know that statistical uncertainties are not the leading challenge for global SMEFT analyses. So if the correlations between experimental systematics are not really relevant either, which uncertainties actually dominate the SMEFT analysis? While for the Higgs sector, the modeling of theory uncertainties has surprisingly little effect on the SMEFT limits [64], the QCD nature of top pair production suggests that the situation will be different here. As a test, in Fig. 12, we repeat the comparison of Fig. 11, neglecting all theory uncertainties. As before, we show the global analysis with correlated systematics in blue, while in black these correlations are removed. Now we see a significant difference. When neglecting the correlations, we observe an increase in the size of the constraints as well as a sizeable shift in the most likely point. This is particularly marked in the 2-dimensional constraints on $C_{Qq}^{18}$ and $C_{tq}^8$.

Comparing Figs. 11 and 12 we learn the importance of theory uncertainties in the top sector. If we neglect the theory uncertainties, the effect of correlated systematics is non-negligible. While theory uncertainties currently wash out these effects, we expect them to become more important as SM calculations become more precise, and theory uncertainties are reduced.

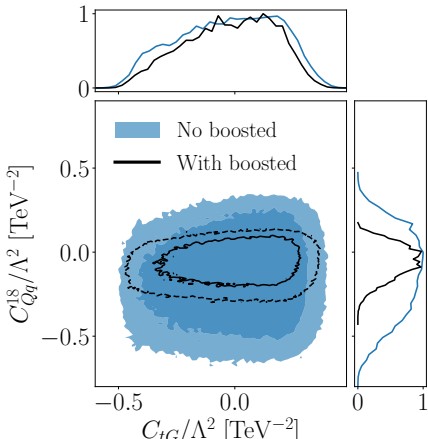
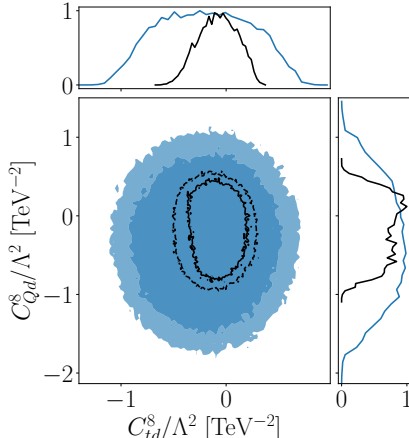

Figure 10: Profile likelihood correlations showing the impact of boosted top pair kinematics [93] (black), compared to the same dataset without this one measurement (blue).

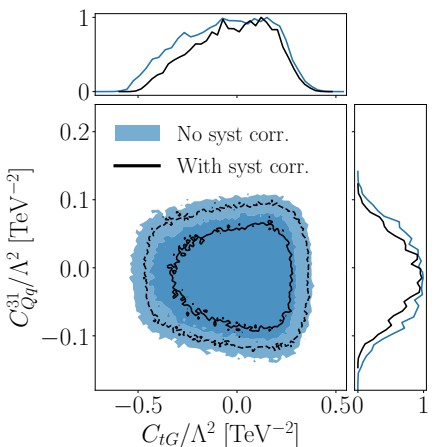
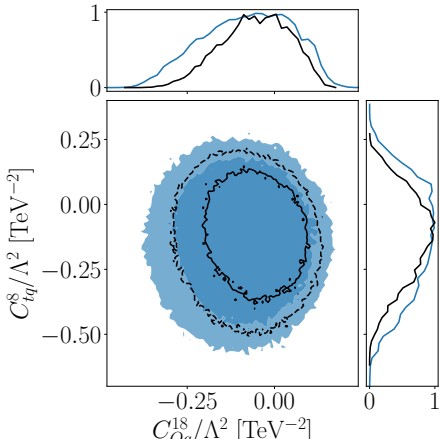

Figure 11: Profile likelihood correlations including correlated systematic and theory uncertainties (blue) versus ignoring correlations between experimental systematics (black).

Moreover, we cannot make any statement about the potential impact of public likelihoods for those kinematic measurements that drive the SMEFT sensitivity.

## 4.3 Marginalization vs profiling

As defined in Eq.(8), the central object of any SFITTER analysis is a fully exclusive likelihood. It is evaluated over the combined space of Wilson coefficients and nuisance parameters. Obviously, the nuisance parameters are irrelevant to the physics interpretation of the global SMEFT analysis. In addition, we are usually not interested in showing all 22 Wilson coefficients at the same time and instead reduce this space to one or two dimensions. Statistically, this can be done by profiling or marginalizing the likelihood. Only for perfect Gaussian distributions do the two methods give the same results, as discussed in Sec. 2.3. In the Higgs-electroweak sector, significant deviations between the two methods appear through a large under-fluctuation in one bin of a kinematic distribution [64].

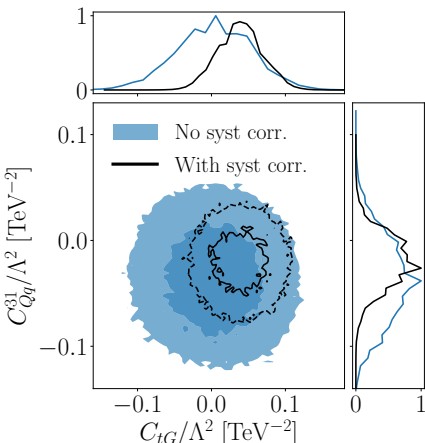
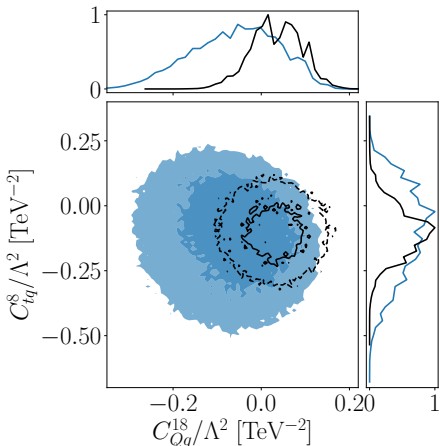

Figure 12: Profile likelihood correlations, ignoring theory uncertainties altogether, and either including correlated systematic uncertainties (blue) or ignoring correlations between experimental systematics (black).

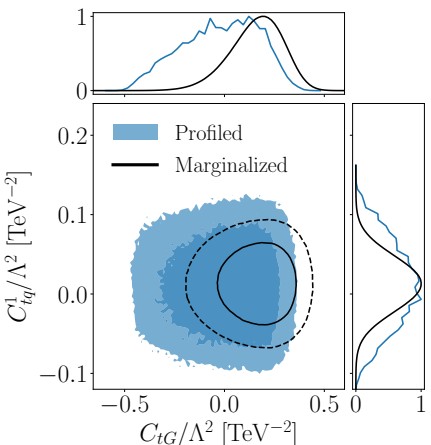
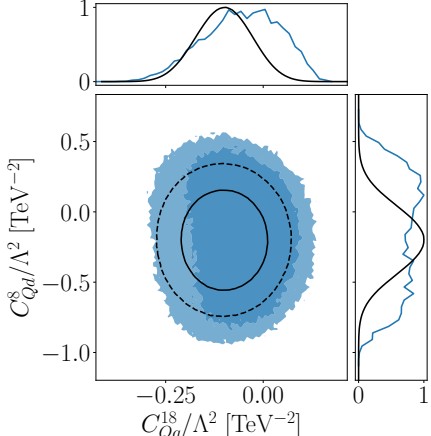

Figure 13: Comparison between marginalization (black) and profiling (blue) in a global analysis of the top sector.

Fig. 13 displays a selection of correlations from a marginalization (black) and profiling (blue) of the fully exclusive likelihood from all top sector measurements and Wilson coefficients. We show constraints for $\mathcal{O}_{tG}$ vs $\mathcal{O}_{tq}^1$ and $\mathcal{O}_{Qq}^{18}$ vs $\mathcal{O}_{Qd}^8$, but similar effects can be seen in many operator pairs. In general, marginalization leads to narrower constraints than profiling. This is particularly evident in the left panel of Fig. 13, and it is due to theory uncertainties and their flat likelihood distribution. With this choice, the profile likelihood can force a perfect agreement between data and predictions over a wide range of values for critical Wilson coefficients. When we marginalize over the exclusive likelihood, the difference between Gaussian and flat uncertainties is less pronounced, leading to more Gaussian and narrower one-dimensional distributions, as discussed in detail in Ref. [64]. This effect is especially visible in the top sector, where theory uncertainties are not only poorly defined [137], but also large.

## 4.4 Top-Higgs-electroweak combination

Finally, making use of the numerical improvements in the SFITTER implementation, we can combine the top-sector SMEFT analysis from this paper with the SFITTER analysis of the Higgs, di-boson, and electroweak precision observables, Ref. [64]. This combination has been stud-

ied in the literature in detail, showing that the two sectors are linked, for instance, through $\mathcal{O}_{tG}$ [35, 36].

We confirm this state of the art and show the combined SFITTER profile likelihood of the two sectors in Fig. 14. In total, 43 degrees of freedom are constrained: the 22 coefficients constrained by the top sector and discussed in Sec. 2.1, and 21 additional operators relevant to the Higgs, di-boson and electroweak observables. The notation and conventions for these 21 operators are provided in App. B. From the detailed discussion above and in Ref. [64], it is clear that the challenges and limitations of the global analyses in the two sectors are not the same. We show the limits at 95% CL from one-dimensional profile likelihoods of the combined fit (blue) and under the assumption of theory uncertainties reduced by a factor of 2 (orange). The numerical values of the constraints shown in Fig. 14 are provided in Tab. 6.

In the top sector, we find strong constraints on the four-fermion operators. The constraints on their Wilson coefficients are driven by kinematic distributions such as the ATLAS measurement of boosted top discussed in Sec. 2.2, and therefore theory uncertainties do not play an important role in their constraints. Conversely, the constraint on $C_{tG}$ improves significantly when theory uncertainties are halved, indicating that theory uncertainties dominate constraints obtained from top quark pair production total cross sections. Similarly, this hypothetical reduction of theory uncertainties has an effect on the constraints obtained from single top, $t\bar{t}W$ and $t\bar{t}Z$ on coefficients such $C_{tW}$, $C_{bW}$, and $C_{tZ}$.

On the other hand, we observe no significant changes in the constraints on the operators relevant to the Higgs, di-boson and electroweak sectors, shown in the lower half of Fig. 14, when theory uncertainties are reduced. The exception is $C_{\phi G}$, which also benefits from the top quark data through its correlation with $C_{tG}$ and $C_{t\phi}$. This is in agreement with Ref. [64], where it was found that in the Higgs-gauge sector, systematic uncertainties are the dominant source of uncertainty for many of the observables in this sector.

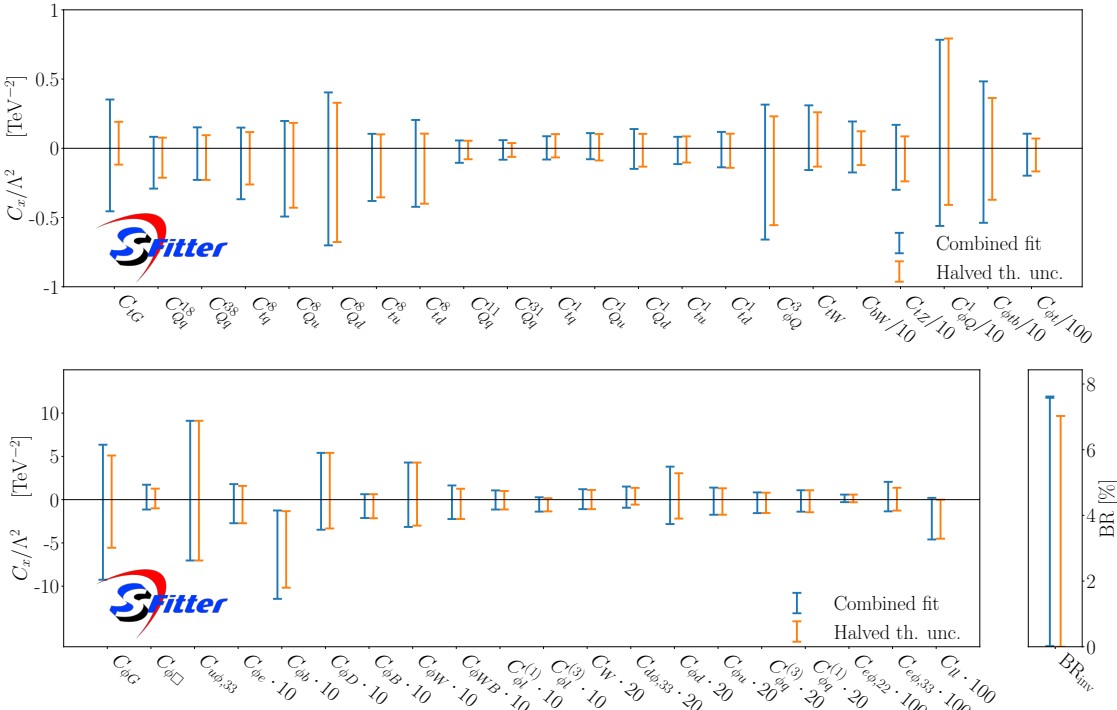

Figure 14: Results from a combined SMEFT analysis of the top sector and the Higgs-electroweak sector, showing the constraints at 95% CL on 43 degrees of freedom, resulting from a profiled likelihood.

# 5 Outlook

Global SMEFT analyses are an exciting development at the LHC, as they combine their role as a precision hadron collider with the goal of interpreting all measurements in terms of precision quantum field theory. This precision theme implies that even if we know that the current measurements do not rule out the Standard Model, limits on SMEFT Wilson coefficients tell us important information about fundamental physics.

To extract limits on fundamental physics parameters, we need a comprehensive uncertainty treatment covering experimental statistical uncertainties, experimental systematics, and theory uncertainties. For the latter two, it is crucial that we include correlations. Public likelihoods are the state of the art in communicating such experimental results to a broader community. We include, for the first time, public ATLAS likelihoods for cross section measurements in a global analysis. These public likelihoods allow us to systematically evaluate the effects of correlations of systematic and theory uncertainties on a global analysis.

The basis of the global SFITTER analysis is a fully exclusive likelihood. It includes a large set of rate and kinematic measurements, either pre-processed by ATLAS or CMS, unfolded, or extracted and backward-engineered from experimental publications. The uncertainty treatment is especially flexible, including a choice of flat nuisance parameters for correlated theory uncertainties. Starting from the fully exclusive likelihood, we can employ a profile likelihood or a Bayesian marginalization to extract limits on individual Wilson coefficients. In the top sector, we find no significant difference between the two statistical approaches.

The focus of this paper was on the role of different uncertainties, their correlations and the role of public likelihoods in this context. In a similar analysis, albeit without public likelihoods, we found that in the electroweak sector, the correlations were crucial, whereas the theory uncertainties were not (yet) a limiting factor [64]. Intriguingly, the situation in the top sector is the opposite: theory uncertainties are crucial, while the correlations of experimental systematics have a limited impact on the SMEFT limits. This reflects the QCD nature and the vast statistics of top pair production.

We have demonstrated that public likelihoods provide a much more flexible approach to handling nuisance parameters. However, fully leveraging their potential currently proves difficult due to the large number of measurements included in our global analysis. We emphasize that this is not a final statement about public likelihoods in SMEFT analyses. The reason is that we find kinematic measurements of boosted top pair production to be the driver behind improved SMEFT limits. For unfolded kinematic measurements, there are no public likelihoods available yet, but we are looking forward to implementing them in SFITTER in the future.

We finished this study of the impact of theory uncertainties in a consistent theory framework of LHC data by performing the first combined SFITTER analysis of the Higgs, electroweak, and top sectors. This further displayed the limiting effect of theory uncertainties on the constraining power of modern top measurements compared to those in the Higgs sector.

## Acknowledgments

We would like to thank Dirk Zerwas, James Moore, and Luca Mantani for discussions on the challenges of Monte Carlo toys, as well as Sabine Kraml and Lukas Heinrich for the encouragement to use public likelihoods. We are also grateful to Luca Mantani for his help with the calculation of SMEFT predictions and Tomas Dado for his hands-on help with public likelihoods.

**Funding information** This research is supported by the Deutsche Forschungsgemeinschaft (DFG, German Research Foundation) under grant 396021762 – TRR 257 *Particle Physics Phenomenology after the Higgs Discovery*. NE is funded by the Heidelberg IMPRS *Precision Tests of Fundamental Symmetries*. MM acknowledges support from the Alexander von Humboldt Foundation. We acknowledge support by the state of Baden-Württemberg through bwHPC and the German Research Foundation (DFG) through grant no INST 39/963-1 FUGG (bwForCluster NEMO).

# A Toys vs Markov chain

The current SFITTER methodology relies on a Markov chain to encode the fully exclusive likelihood given in Eq.(8). It then allows for a profile likelihood or marginalization to remove nuisance parameters and extract limits on individual Wilson coefficients [62, 64]. Past SFITTER analyses used an alternative methodology known as Monte Carlo toys [11,62,63] or Monte Carlo replicas [20,38,133,146]. This method has not been used in SFITTER since the SMEFT analysis of the top sector [21]. In Ref. [38], its shortcomings for a SMEFT analysis of the top sector are discussed in detail. In this appendix, we provide an additional discussion of Monte Carlo toys in the SFITTER context.

**Likelihood**

The basis of, essentially, all LHC analyses is the likelihood of a given measurement, $d$, compared to a model or theory prediction $m(c)$, which depends on parameters or Wilson coefficients $c$ [64]. To simplify this discussion, we approximate it as a Gaussian,

$$\log p(d|c) = -\frac{[d - m(c)]^2}{2\delta^2} = -\frac{\chi^2(c)}{2}, \tag{A.1}$$

with an uncertainty $\delta$ assigned to the measurement.

The toys method describes possible outcomes of a measurement, given its uncertainty, with a nuisance parameter. In the Gaussian limit, the outcome of a measurement is sampled around the mean $\bar{d}$,

$$d_k \sim \mathcal{N}(\bar{d}, \delta). \tag{A.2}$$

To extract the likelihood in Eq.(A.1) we generate a set of toy-measurements $d_k$, mimicking the outcomes of actual measurements. The basic frequentist assumption is that we can maximize the likelihood for each toy-measurement,

$$c_k = \text{argmax } p(d_k|c), \tag{A.3}$$

to extract the maximum-likelihood parameters for each outcome, and infer the likelihood over model space from the density of these points $\{c_k\}$.

We can compare the toys to a simple Markov chain, where we are only interested in a sample of points representing a given likelihood distribution, without any downstream task. At each step, the Markov chain proposes a new parameter point and keeps it with the probability defined by the likelihood in Eq.(A.1). As for the toys, the Markov chain encodes the likelihood through a density of points. The difference is that the toys start from the distribution of the experimental measurements and extract the likelihood from maximum-likelihood points, while the Markov chain collects points proportionally to the likelihood. Both methods give the same results, provided their algorithms respect the assumption that the distribution of maximum-likelihood points reproduces the underlying probability or likelihood.

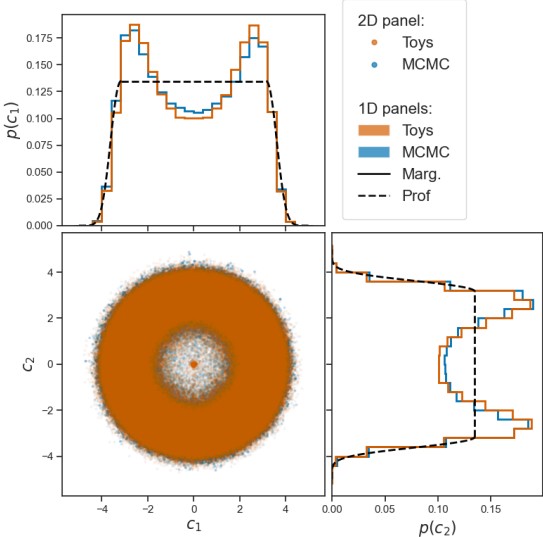

Figure 15: Comparison between toys and MCMC, constraining $c_1$ and $c_2$ using the measurement $d_1$ from Eq.(A.5). The 1-dimensional distributions show the profile likelihoods, as well as the marginalized probabilities. The profiling is obtained by analytically maximizing the log-likelihood of Eq.(A.1) along each axis.

An interesting challenge is the description of correlations between measurements in the likelihood. For the Gaussian case, we can describe them with a correlation matrix if the correlation is less than 100%. To describe correlations for non-Gaussian likelihoods, we introduce nuisance parameters, which increase the effective dimensionality of the likelihood.

We have introduced toys and the Markov chains as ways to construct a likelihood $p(d|c)$ for a given dataset $d$. We can as well introduce them as tools to construct the probability map $p(c|d)$. If the prior $p(c)$ entering the two algorithms is constant over a wide enough range, the distribution of points will describe the likelihood as well as the probability $p(c|d)$, the only difference being a normalization constant.

**Circular flat direction**

A key feature of the SMEFT analysis in the top sector is that, typically, 4-fermion operators have extremely small interferences with the SM. The likelihood as a function of the Wilson coefficients is dominated by the squares of these coefficients. A measurement constraining two different Wilson coefficients could then read

$$d \sim m(c) = m_{\text{SM}} + 0.1c_1^2 + 0.1c_2^2 \,. \tag{A.4}$$

If we consider $c_i^2$ the model parameters, we can always find two squared Wilson coefficients which solve this relation. In contrast, if we consider $c_i$ as model parameters, we can only solve it if the measurement is an upward fluctuation, $d > m_{\text{SM}}$. An example would be

$$d_1 = 6 > m_{\text{SM},1} = 5 \,, \qquad \text{solved by} \qquad 0.1c_1^2 + 0.1c_2^2 = 1 \,. \tag{A.5}$$

We show the correlated values of $c_1$ and $c_2$ in Fig. 15. The Markov chain and the toys form the same circle. The width of this circular flat direction is given by $\delta_{1,2} = 0.3$. The 2-dimensional likelihood is extracted by binning, with Markov chains and toys producing the same result.

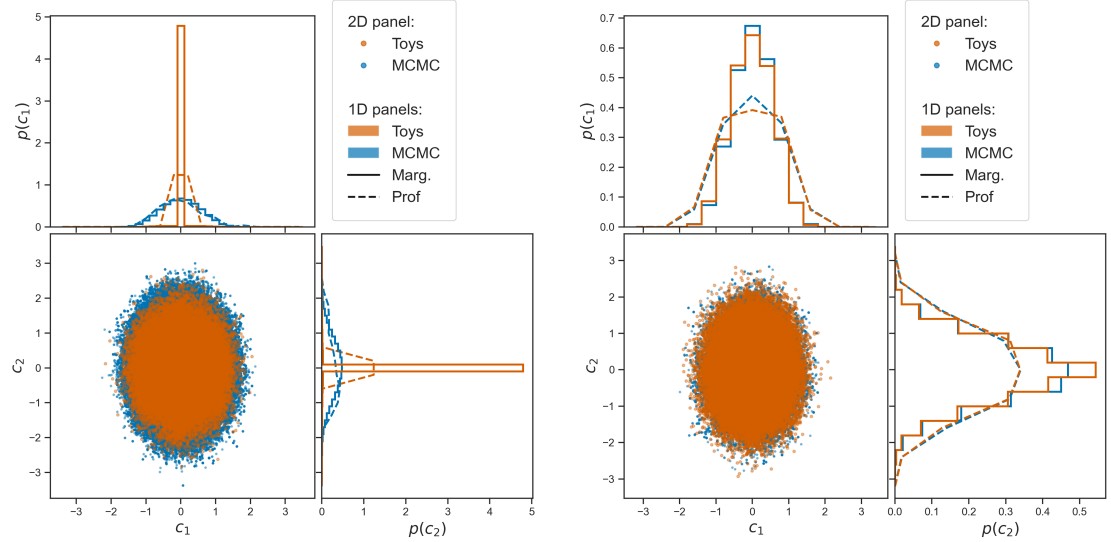

Figure 16: Left: comparison between toys and MCMC, constraining $c_1$ and $c_2$ using the measurement $d_2$ from Eq.(A.6). Right: the same comparison, but removing the peak of toys at the boundary $c_{1,2} \approx 0$. The 1-dimensional distributions show the profile likelihoods as well as the marginalized probabilities.

The 2-dimensional likelihood can now be reduced in dimensionality. Here, profiling and marginalization lead to differences through volume effects [64], as seen in the 1-dimensional distributions in Fig. 15. This difference is independent of the toys and Markov chain, which completely agree.

**Unexplainable underfluctuation**

A problem occurs when we encounter a negative fluctuation in the measurement, which cannot be mapped to the usual minimum in model space. For instance,

$$d_2 = 6 < m_{\text{SM},2} = 7, \qquad \text{requiring} \qquad 0.2c_1^2 + 0.1c_2^2 \overset{!}{=} -1. \qquad (A.6)$$

This relation cannot be solved for real $c_i$, and the spread from $\delta_{1,2} = 0.3$ is too small to cure this problem. The likelihoods obtained from toys and from the Markov chain are both given in the left panel of Fig. 16, and they differ. The Markov chain includes points with a finite likelihood offset below the theoretical maximum. The toys are derived from Eq.(A.3), returning $c_1 = c_2 = 0$ if they cannot reach the true maximum. For the right panel of Fig. 16, we modify the toys algorithm to remove the maximum-likelihood peak at the parameter space boundary. This is done by retaining only the samples $d_k$ in Eq.(A.2) which satisfy $d_k > m_{\text{SM},2}$. With this modification, the Markov chain and the toys have the same likelihood.

Summarizing Fig. 16, the toys and the Markov chain treat the unwanted parameter space $c_i^2 < 0$ differently. While the toys provide a maximum-likelihood parameter point for each assumed measurement, the Markov chain removes the unwanted points. In terms of a prior on the numerical implementation of the scanning, we can understand the two methods as

$$
\begin{aligned}
\text{toys:} \qquad & p(d|c)\Big|_{\text{phys}} \sim p(c|d)\Big|_{\text{phys}} = \max(\Delta, p(c|d)), \\
\text{Markov chain:} \quad & p(c) = \Theta(c^2),
\end{aligned}
\qquad (A.7)
$$

where $\Delta$ is chosen to remove the numerically broadened peak at $c = 0$. In the right panel of Fig. 16 we see that for allowed $c_i$ the two methods give the same result but with a different



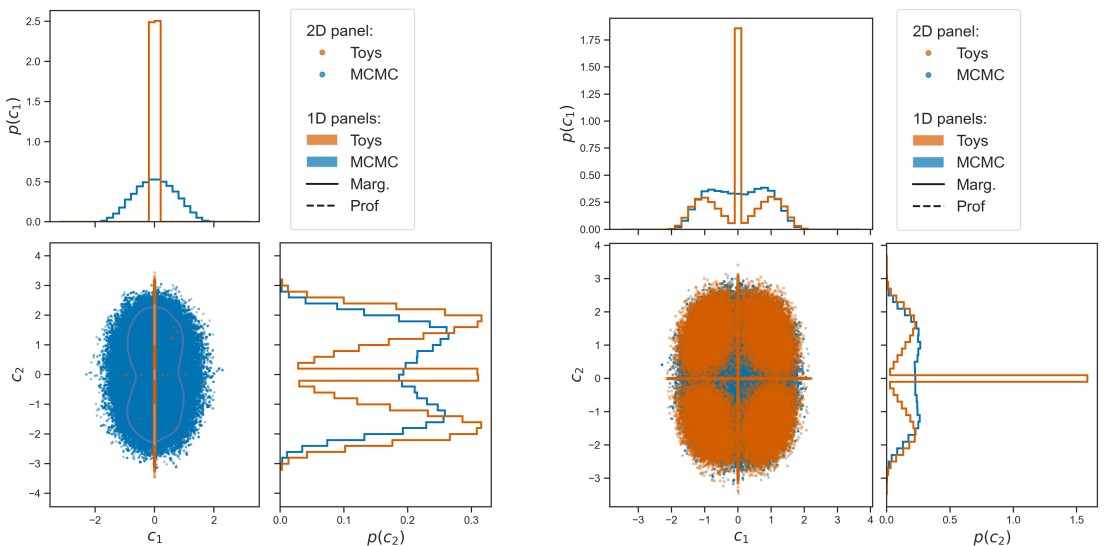

Figure 17: Left: comparison between toys and MCMC for the combination of $d_1$ and $d_2$ in terms of the $c_i$. Right: the same for a combination of $d_1$, $d_2$, and $d_3$.

normalization of the probability $p(c|d)$ [38]. If we work with the likelihood $p(d|c)$, the normalization does not matter. If we include all toy experiments, the two distributions in Fig. 16 reflect the fact that the two methods are asking different statistical questions.

**Spreading out the peak**

The situation given by Eq.(A.6) is just an extremely unlucky outcome. Of very few measurements, one happens to be many standard deviations away from the prediction, while we expect such an outlier only once in many more measurements.

Moreover, in a realistic global analysis, many measurements together constrain a given model parameter. To see what happens then, we first combine $d_1$ and $d_2$ in the left panel of Fig. 17. The Markov chain generates a valid point distribution, symmetric in $\pm|c_2|$. For the toys, the situation is different. On the underfluctuation in $d_2$, $c_1$ and $c_2$ have the same impact, but for the overfluctuation in $d_1$, a shift in $c_2$ gains more. This is why a peak is observed at $c_1 = 0$ and $c_2$ is adjusted.

Next, we add a third measurement, such that the effect of our underfluctuation will be compensated by another pull on $c_1$,

$$d_3 = m_{\text{SM},3} + 0.2c_1^2, \qquad \text{with} \qquad d_3 = 6, \, m_{\text{SM},3} = 5.5. \qquad (A.8)$$

The central maximum of the corresponding likelihood will be at $c_1 \approx 1.6$. In the right panel of Fig. 17, we see that the peak at $c_1 = 0$ is now accompanied by a distribution of finite $c_1$ symmetric around zero. This way, the importance of the peak is reduced. For more measurements, this will continue until the underfluctuation will just be an expected statistical outlier in a large set of measurements, with little effect on the global likelihood.

**Determining squared coefficients**

Because our measurements show a purely quadratic dependence, we can try to circumvent our problem by extracting the likelihood over a model space defined by $c_1^2$ and $c_2^2$. Instead of first requiring $c_i^2$ to be positive and then maximizing the log-likelihood, we first maximize the log-likelihood in terms of the $c_i^2$. In the left panel of Fig. 18, we show the results in terms of

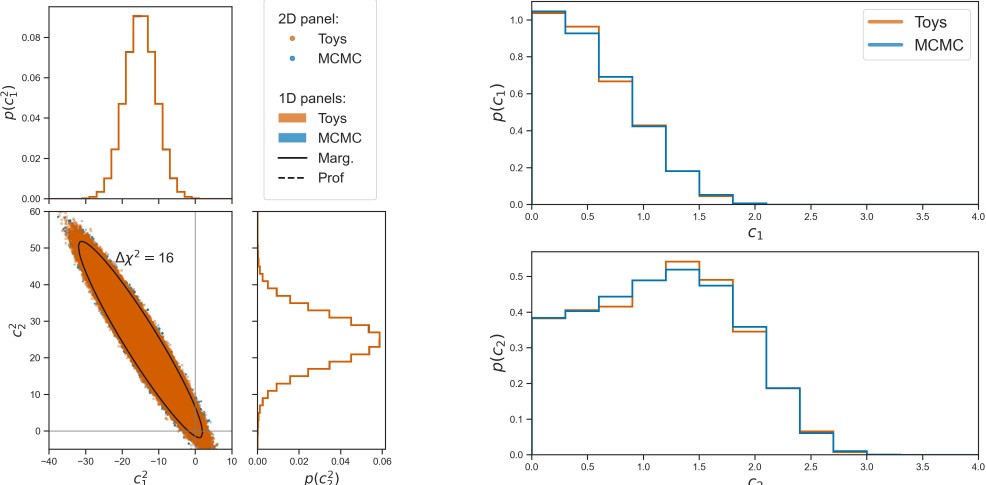

Figure 18: Comparison between toys and MCMC for the combination of $d_1$ and $d_2$ in terms of the $c_i^2$, and for the valid points only in terms of the $c_i$.

the $c_i^2$, with toys and Markov chain in perfect agreement. However, almost the entire preferred range gives unwanted values for $c_2^2$. The curve defined by $\Delta\chi^2 = 16$ highlights how unlikely it is to obtain points with $c_i^2 > 0$.

In the second step, we now select valid parameter points. This avoids a hard boundary when optimizing the toys. In the right panel of Fig. 18, the blue curves show the points in the Markov chain, sampling directly in $c_1$ vs $c_2$. These are the same points as those in the upper-right quadrant of the left panel of Fig. 17. In orange, we show the one-dimensional distributions of only the allowed parameter points out of all those plotted in the left panel of Fig. 18. Neither the likelihood nor the probability is parametrization invariant, so we need to apply the Jacobian

$$p(c) = p(c^2) \frac{dc^2}{dc}, \tag{A.9}$$

when extracting the shown likelihoods as a function of $c_i$ rather than $c_i^2$. For comparison with the blue curves, the orange distribution is reweighted by this Jacobian. Again, we find complete agreement between the toys and the Markov chain.

## B Higgs, top, di-boson and electroweak combination

In producing the global analysis of Sec. 4.4 and Fig. 14, we have combined the top sector from this paper with the previous SFITTER analysis of the Higgs, di-boson, and electroweak sectors of Ref. [64], taking all data from within this reference. Note that while Ref. [64] provides constraints on Wilson coefficients in the HISZ basis; here, we provide all constraints in the Warsaw basis.

In addition to the 22 operators introduced in Sec. 2.1 and constrained by the top sector observables, a further 21 Wilson coefficients can be constrained by the addition of data from the Higgs, di-boson, and electroweak sectors. These operators are assumed to follow the same flavor symmetry conventions as introduced in Eq.(2), *i.e.* flavor universality applied to the first two quark generations. The notation and conventions for these 21 operators are provided in Tab. 5.

Table 6 reports the numerical values of the boundaries of the 95% CL intervals shown in Fig. 14.

Table 5: Additional Wilson coefficients of the Warsaw basis entering the combined analysis of the Higgs, di-boson, top, and electroweak sectors. The 21 degrees of freedom shown here are included in the global analysis of Sec. 4.4 alongside the 22 operators already constrained by top sector observables. In total, 43 coefficients are constrained in the global analysis.

| Coefficient | Operator | Coefficient | Operator |
|---|---|---|---|
| $C_{\phi G}$ | $\phi^\dagger \phi\, G^A_{\mu\nu} G^{A\mu\nu}$ | $C_W$ | $\varepsilon^{IJK} W^{I\nu}_\mu W^{J\rho}_\nu W^{K\mu}_\rho$ |
| $C_{\phi\Box}$ | $(\phi^\dagger \phi)\Box(\phi^\dagger \phi)$ | $C_{d\phi,33}$ | $(\phi^\dagger \phi)(\bar{Q}_3 b \phi)$ |
| $C_{u\phi,33}$ | $(\phi^\dagger \phi)(\bar{Q}_3 t \phi)$ | $C_{\phi d}$ | $\sum_{i=1}^2 (\phi^\dagger i \overleftrightarrow{D}_\mu \phi)(\bar{d}_i \gamma^\mu d_i)$ |
| $C_{\phi e}$ | $(\phi^\dagger i \overleftrightarrow{D}_\mu \phi)(\bar{e}\gamma^\mu e)$ | $C_{\phi u}$ | $\sum_{i=1}^2 (\phi^\dagger i \overleftrightarrow{D}_\mu \phi)(\bar{u}_i \gamma^\mu u_i)$ |
| $C_{\phi b}$ | $(\phi^\dagger i \overleftrightarrow{D}_\mu \phi)(\bar{b}\gamma^\mu b)$ | $C^{(3)}_{\phi q}$ | $\sum_{i=1}^2 (\phi^\dagger i \overleftrightarrow{D}_\mu \phi)(\bar{q}_i \gamma^\mu q_i)$ |
| $C_{\phi D}$ | $(\phi^\dagger D^\mu \phi)^*(\phi^\dagger D^\mu \phi)$ | $C^{(1)}_{\phi q}$ | $\sum_{i=1}^2 (\phi^\dagger i \overleftrightarrow{D}_\mu \phi)(\bar{q}_i \tau^I \gamma^\mu q_i)$ |
| $C_{\phi B}$ | $\phi^\dagger \phi\, B_{\mu\nu} B^{\mu\nu}$ | $C_{e\phi,22}$ | $(\phi^\dagger \phi)(\bar{l}_2 \mu \phi)$ |
| $C_{\phi W}$ | $\phi^\dagger \phi\, W^I_{\mu\nu} W^{I\mu\nu}$ | $C_{e\phi,33}$ | $(\phi^\dagger \phi)(\bar{l}_3 \tau \phi)$ |
| $C_{\phi WB}$ | $\phi^\dagger \tau^I \phi\, W^I_{\mu\nu} B^{\mu\nu}$ | $C_{ll}$ | $(\bar{l}\gamma_\mu l)(\bar{l}\gamma^\mu l)$ |
| $C^{(1)}_{\phi l}$ | $(\phi^\dagger i \overleftrightarrow{D}_\mu \phi)(\bar{l}\gamma^\mu l)$ | $BR_{inv}$ | invisible Higgs decays |
| $C^{(3)}_{\phi l}$ | $(\phi^\dagger i \overleftrightarrow{D}^I_\mu \phi)(\bar{l}\tau^I \gamma^\mu l)$ | | |

Table 6: Numerical values for the 95% CL limits shown in Fig. 14. We emphasize that the reduction of the theory uncertainties by a factor two is entirely hypothetical.

| Coefficient | Full analysis | Halved theory unc. | Coefficient | Full analysis | Halved theory unc. |
|---|---|---|---|---|---|
| $C_{\phi G}$ | [-9.25, 6.35] | [-5.56, 5.1] | $C_{tG}$ | [-0.46, 0.35] | [-0.12, 0.19] |
| $C_{\phi \Box}$ | [-1.14, 1.72] | [-1.0, 1.27] | $C_{Qq}^{(18)}$ | [-0.29, 0.08] | [-0.21, 0.08] |
| $C_{u\phi,33}$ | [-7.03, 9.11] | [-7.03, 9.11] | $C_{Qq}^{(38)}$ | [-0.23, 0.15] | [-0.23, 0.09] |
| $C_{\phi e} \times 10$ | [-2.73, 1.79] | [-2.73, 1.59] | $C_{tq}^{(8)}$ | [-0.37, 0.15] | [-0.26, 0.12] |
| $C_{\phi b} \times 10$ | [-11.46, -1.25] | [-10.17, -1.33] | $C_{Qu}^{(8)}$ | [-0.49, 0.2] | [-0.43, 0.18] |
| $C_{\phi D} \times 10$ | [-3.48, 5.4] | [-3.33, 5.4] | $C_{Qd}^{(8)}$ | [-0.7, 0.4] | [-0.68, 0.33] |
| $C_{\phi B} \times 10$ | [-2.12, 0.63] | [-2.15, 0.63] | $C_{tu}^{(8)}$ | [-0.38, 0.1] | [-0.35, 0.1] |
| $C_{\phi W} \times 10$ | [-3.15, 4.29] | [-3.0, 4.28] | $C_{td}^{(8)}$ | [-0.42, 0.2] | [-0.4, 0.11] |
| $C_{\phi WB} \times 10$ | [-2.24, 1.64] | [-2.24, 1.26] | $C_{Qq}^{(11)}$ | [-0.1, 0.06] | [-0.08, 0.05] |
| $C_{\phi l}^{(1)} \times 10$ | [-1.14, 1.07] | [-1.14, 1.0] | $C_{Qq}^{(31)}$ | [-0.08, 0.06] | [-0.06, 0.04] |
| $C_{\phi l}^{(3)} \times 10$ | [-1.38, 0.27] | [-1.36, 0.17] | $C_{tq}^{(1)}$ | [-0.08, 0.09] | [-0.07, 0.1] |
| $C_W \times 20$ | [-1.1, 1.2] | [-1.1, 1.12] | $C_{Qu}^{(1)}$ | [-0.08, 0.11] | [-0.09, 0.1] |
| $C_{d\phi,33} \times 20$ | [-0.94, 1.51] | [-0.58, 1.36] | $C_{Qd}^{(1)}$ | [-0.15, 0.14] | [-0.13, 0.1] |
| $C_{\phi d} \times 20$ | [-2.83, 3.81] | [-2.19, 3.05] | $C_{tu}^{(1)}$ | [-0.11, 0.08] | [-0.1, 0.09] |
| $C_{\phi u} \times 20$ | [-1.75, 1.39] | [-1.75, 1.31] | $C_{td}^{(1)}$ | [-0.14, 0.12] | [-0.14, 0.11] |
| $C_{\phi q}^{(3)} \times 20$ | [-1.56, 0.84] | [-1.54, 0.8] | $C_{\phi Q}^{(3)}$ | [-0.66, 0.32] | [-0.56, 0.23] |
| $C_{\phi q}^{(1)} \times 20$ | [-1.39, 1.08] | [-1.46, 1.08] | $C_{tW}$ | [-0.16, 0.31] | [-0.13, 0.26] |
| $C_{e\phi,22} \times 100$ | [-0.29, 0.58] | [-0.3, 0.58] | $C_{bW}/10$ | [-0.17, 0.19] | [-0.12, 0.12] |
| $C_{e\phi,33} \times 100$ | [-1.35, 2.06] | [-1.26, 1.37] | $C_{tZ}/10$ | [-0.3, 0.17] | [-0.24, 0.09] |
| $C_{ll} \times 100$ | [-4.61, 0.21] | [-4.51, 0.0] | $C_{\phi Q}^{(1)}/10$ | [-0.56, 0.78] | [-0.41, 0.79] |
| $BR_{inv}$ | [0, 7.6] | [0, 7.03] | $C_{\phi tb}/10$ | [-0.54, 0.48] | [-0.37, 0.36] |
| | | | $C_{\phi t}/100$ | [-0.2, 0.11] | [-0.17, 0.07] |

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
