# Peer review of "Staying on Top of SMEFT-Likelihood Analyses"

_SciPost Physics, doi:SciPost Phys. 18, 108 (2025)_

## Round 2 · Referee Report · Anonymous (Referee 2) · 2024-7-16

Strengths

1 - first usage of public likelihoods
2 - study of the impact of systematic uncertainties
3 - careful study of the reproducibility of the results from externally provided inputs

Weaknesses

1 - The language, overall, should be improved
2 - Too few details on the concrete model and the uncertainty implementation in the combination
3 - The results from the combined SMEFT analyses should be better integrated into the narrative
4 - A question about the validity of the statistical procedure related to the Markov chain should be addressed

Report

The paper draft fits the journal's scope.

Requested changes

Please see the attached text file. Since several suggestions can potentially change the results, I mark it as a "major revision".

Attachment

Recommendation

Ask for major revision

  • validity: top
  • significance: top
  • originality: top
  • clarity: top
  • formatting: good
  • grammar: good

Author:  Nina Elmer  on 2025-01-09  [id 5100]

(in reply to Report 1 on 2024-07-16)
Category:
answer to question
correction

We thank the referee for the careful revision of our manuscript. Here, we address the various points raised in their reports and we explain how the manuscript has been modified.

In many places, I found the language somewhat vague and colloquial, bordering jargon. A revised version would profit from a leaner and more crisp language. → We had a careful look at the language and changed it whenever we identified jargon.

I found the description of the construction of the likelihood (p9) too vague. → We changed the description whenever we thought additional information was helpful and added a comment on flat likelihoods.

More detailed comments 1) Table 2: There are several potentially important measurements not included. ((very new) total+differential tt/tt+jets in lepton+jets ATLAS https://arxiv.org/abs/2406.19701, total+differential ttW ATLAS https://arxiv.org/abs/2401.05299, total+differential ttW CMS https://arxiv.org/abs/2208.06485, tt+gamma 13 TeV ATLAS https://arxiv.org/abs/1812.01697 https://arxiv.org/abs/2403.09452, tt+gamma 13 TeV CMS https://arxiv.org/abs/2107.01508 https://arxiv.org/abs/2201.07301 The ttbar spin correlation measurement provides tt(2l) unfolded cross sections that could constrain ctG (your Fig 10) better than the combination. → We agree that some of these analyses might be useful, but especially the differential tt+jets measurement was published long after the original version of the paper. Total rates and/or low-rate associated channels are unlikely to affect our analysis much. We decided to stick to the analyses included in the original paper and will consider these analyses in our next, dedicated SFitter top sector analysis.

2) p7/8, Fig. 1. I found the provided examples of the SMEFT sensitivity relatively short. It is not possible to show all possible variations, but a few more examples, possibly with other operators would be very interesting. Why not put all the inclusive cross-sections in a figure, one in each bin, and show how these change with SMEFT by overlaying a few selected parameter points? → We are sorry, but we are not sure what the referee is referring to. Fig.1. illustrates the effect of one especially interesting operator on a kinematic distribution. That is really the only point we feel we need to discuss further, more information was included in our comprehensive analysis 1910.03606.

3) You emphasize correctly in the Appendix that the Monte-Carlo replica method has shortcomings, related, in particular, to the quadratic SMEFT terms. But in Eq. 12 you use the asymptotic Gaussian assumption for the Markov Chain. This seems to imply Wilk/Wald which could be violated by quadratic terms as discussed in arxiv:2207.01350. The Gaussian approximation could be invalid in this case. For example, Fig. 1 shows positive modifications for positive and negative values for CQd8, indicating that quadratics are dominant at the 3sigma level. It should be checked (or explained) why/how/if the quadratic terms invalidate the statistical theorems. (I believe the tests in, e.g., Sec. 3.1 are linear in the POI and would are not conclusive in this regard.) → We use the Gaussian approximation for the statistical part of the SFitter likelihood for the top sector, because it simplifies things, and because we convinced ourselves that, unlike in the Higgs sector, we do not have to use the general Poisson form. This affects only the construction of the fully exclusive likelihood, the marginalization and the profiling are discussed in the Appendix. We have not found any fundamental issues with the validity of these two approaches, once we keep in mind that they can give different results, for instance in the presence of flat theory uncertainties. This means we are not confident to comment on fundamental violations of statistical theorems through, for us, just numerical simplifications. We don’t use toys in our SFitter implementation

4) I think Section 2.3 can be improved in terms of the quality of the text and the level of detail in the construction of the likelihood. In particular, p9 and the paragraph starting with "By ansatz..." can be significantly improved. It is not quite clear how theory uncertainties are correlated. Even though the treatment is heuristic, they should be correlated across bins/measurements for each process separately. There is too little detail provided on how the theory uncertainties are obtained (which generator scale choice?). → We went through the text again carefully and adjusted the text where necessary. We provide some additional information on the construction of the likelihood and added a sentence on theory uncertainties and their correlation across different bins and measurements. Furthermore, additional details on how scale uncertainties are obtained have been included.

5) (Section 3) I have no substantial concerns with the studies of the public likelihood. However, the nuisance parameters appearing in the 2D distributions and the naming of the uncertainties in the impact plots should be harmonized. The uncertainties should also be explained briefly in the text. → Only the names for the parameters in the impact plots are provided and are used to allow for an easy comparison. The 2D distributions show nuisance parameters where these names are not provided by experiment so we cannot harmonize these properly. We included a brief comment on this in the text.

6) (Section 4.4) This section should be improved and better integrated in the paper. Several operators appear only in the appendix. It emphasizes OtG, but the most sensitive measurements (see above) are not included. It appears the Higgs combination is a late addition to the draft and stands rather alone. As for the fit in the top sector, lists of operators, uncertainties, etc. should be added. → We went through the section again carefully and adapted the text when necessary. A detailed list of operators for the Higgs and diboson sector can be found in Appendix B or in 2208.08454, as mentioned in the paper.

Minor comments: p2, 2nd paragraph: SMEFT is renormalizable, but not in the sense of the SM which requires a fixed and unchanged number of counter terms to all orders. Please clarify. → That is a good point, but since we are truncating terms beyond dimension six, it makes SMEFT renormalizable again, because we have a finite number of operators and counterterms. This truncation is motivated by the fact that contributions beyond dimension six are negligibly small in a global analysis. As we also state in the second paragraph on page 3.

p2, 3rd paragraph. Many more references than [45] (HighTEA, with 3 references itself) would be needed or the sentence should be changed so that it actually refers to what is done in [45]. → When consulting the High-Tea website, they suggest to only cite this one paper when using it. As mentioned in the text, HighTEA is only used for the K-factor calculation of the ttbar production.

p3, Eq. 1: It looks odd to use the same index k (without bounds) to count two different sets of operators. Moreover, the Weinberg operator (dim5) is not mentioned at all - it should be said that it is not relevant. → We now explicitly mention that the Weinberg operator is neglected and make the notation of the sums more clear.

"Because the underlying symmetry structure is an input to the EFT construction" ... Do you want to say that SMEFT operators have well-defined CP properties and therefore you can remove the CP violating sector? Is the "symmetry structure" the discrete SM symmetries? This paragraph should be made clearer in this regard. → Indeed, the EFT expansion starts with an assumed set of fields and symmetries and then includes all allowed operators at a given dimensionality. We clarified this in the text.

Table 1, Eq.2 What about 2-quark-two-lepton operators? Those affect tt+multilepton final states and are not mentioned. →Sorry for being confused. In the top pair signal such operators would be strongly suppressed off-shell contributions. In the top decays additional operators can be included, but the top decays themselves are limited in the momentum flow and therefore not especially sensitive. We clarified this in the text.

Eq. 6 \mp -> \pm should be more appropriate → We changed it to \pm.

Sec. 2.3 3rd paragraph: I do not understand the 2nd sentence. You do truncate the Lagrangian but also the LHC rate predictions at the quadratic level. → We do indeed truncate both the Lagrangian and rate predictions at the quadratic level. We emphasize that the truncation of the Lagrangian removes any dimension 8 effects from these quadratic contributions.

p9 3rd paragraph. "allowed shift ... at no cost in the likelihood" is unclear. I think you refer to an (additive or multiplicative?) nuisance parameter whose impact is inexpensive to compute. → We rephrased the statement in question to make it more clear.

Sec. 4.1 There are several unnecessary forward references in the text, e.g., in the first sentence. The last two paragraphs (in particular the last sentence) in the outlook do not integrate well with the narrative of the rest of the text. → We went carefully through the text again, and removed any forward references we considered unnecessary.

For a better display of the changes regarding your comments and suggestions, we marked them in orange in the attached version of the paper.

Attachment:

diff_MUprrNc.pdf

---

## Round 2 · Referee Report · Anonymous (Referee 1) · 2024-7-19

Strengths

1- The study uses public likelihood information for the first time in a SMEFT interpretation.
2- A rigorous statistical interpretation is performed, using the well-established SFitter tool.
3- The impact of various systematic uncertainties is quantified, helping to identify where the most impartant effects lie in the interpretation of top quark data.

Weaknesses

1- Some of the explanations are a bit too brief, e.g., the actual implementation of the systematic uncertainties for the datasets used and the details of the information used in constructing the HistFactory likelihoods.
2- The public likelihoods should contain enough information to treat the impact of EFT effects in the background yields. It would have been nice to quantify the impact of those.

Report

The paper is short but achieves its intended goal, showing a proof of principle for using public likelihood information in SMEFT analyses. This is something that has been talked about for many years but has not been possible until recently. It will hopefully motivate experimental collaborations to consistently report their results in this way. The in-depth study of systematic uncertainties for top sector SMEFTinterpretations is timely and will be useful to the community. I am satisfied that the paper meets the threshold for publication in SciPost Physics.

Requested changes

1- p4, above equation (6): The 4-fermion operators are described as having a "chirality flip". I would typically reserve that description for scalar and tensor current operators that directly connect left and right-handed fields.
2- p5, before Sec. 2.2. Four-heavy operaators have also been shown to have an impact on ttbar production. While this impact is likely modest, it should be mentioned for context.
3- p6, why is only tt-gamma not inclued at NLO? It should be discussed whether this is due to a technical challenge or limitation, or simply a choice.
4- p7: I don't understand the statement that a minimum of 10%(2%) scale uncertainties are enforced for total rates (differential cross sections). Does this mean that reported scaled unceertainties below this amount are artificially inflated? Please explain this part more clearly and justify the choice.
5- p8, Fig. 1: I recommend placing a lower panel on each subfigure showing the ratio of the SMEFT prediction to the SM, to clearly show therelative impact of the operator.
6- p9/10, before Sec. 3: The treatment of flat likelihoods by the profile likelihood method is discussed here. What does the Bayesian marginlisation do here? I believe it is briefly discussed later on when both methods are actually compared, but since you summarise one here, it would be clearer if the other were also summarised.
7- p10, Tab. 4: The table is not sufficiently explained in the text/caption. Symbols are undefined such as alpha, gamma, lambda which makes the whole discussion/explanation of the systematics modelling unclear.
8- p18, below Fig. 11: The conclusion on correlated systematics is that they have a negligible impact. Yet it is later stated that they cannot be ignored, which is a bit contradictory. Perhaps rephrase this or remove this statement, since it seems like they can be ignored in this case.
9- p19, before Sec. 4.3: Can you say anything about the impact of neglecting correlations between theory uncertainties? Or is this not possible due to the way that you implement them in SFitter? I believe that theory uncertainty correlations are often neglected in other works, so it would be interesting to see their impact, given the fact that theory errors dominate in this study.

Recommendation

Ask for minor revision

  • validity: high
  • significance: good
  • originality: high
  • clarity: good
  • formatting: excellent
  • grammar: excellent

Author:  Nina Elmer  on 2025-01-09  [id 5101]

(in reply to Report 2 on 2024-07-19)
Category:
answer to question
correction

We thank the referee for the careful revision of our manuscript. We address the various points raised in the report and explain the modifications made in the manuscript.

  1. p4, above equation (6): The 4-fermion operators are described as having a "chirality flip". I would typically reserve that description for scalar and tensor current operators that directly connect left and right-handed fields. → Thank you, we changed this to different-chirality currents

  2. p5, before Sec. 2.2. Four-heavy operators have also been shown to have an impact on ttbar production. While this impact is likely modest, it should be mentioned for context. → We now mention the four-heavy operators in the text on page 5.

  3. p6, why is only tt-gamma not included at NLO? It should be discussed whether this is due to a technical challenge or limitation, or simply a choice → At this point this is a technical choice, given the expected reach of this process.

  4. p7: I don't understand the statement that a minimum of 10%(2%) scale uncertainties are enforced for total rates (differential cross sections). Does this mean that reported scaled uncertainties below this amount are artificially inflated? Please explain this part more clearly and justify the choice. → Here we follow our convention as found in 1910.03606. The point is that an unexpectedly small scale variation does not necessarily point to a small theory uncertainty, but to a failure of the scale variation to represent the correct theory uncertainty due to e.g. strong cancellations of these uncertainties in normalized kinematic distributions.

  5. p8, Fig. 1: I recommend placing a lower panel on each subfigure showing the ratio of the SMEFT prediction to the SM, to clearly show the relative impact of the operator. → The ratio was added as lower panel to Fig. 1, showing the relative impact more clearly

  6. p9/10, before Sec. 3: The treatment of flat likelihoods by the profile likelihood method is discussed here. What does Bayesian marginalization do here? I believe it is briefly discussed later on when both methods are actually compared, but since you summarise one here, it would be clearer if the other were also summarised. → Thank you for noticing the missing discussion of the Marginalization for flat likelihoods. We added a brief discussion of the marginalization to the end of Sec. 2.

  7. p10, Tab. 4: The table is not sufficiently explained in the text/caption. Symbols are undefined such as alpha, gamma, lambda which makes the whole discussion/explanation of the systematics modelling unclear. → A more in depth explanation of the Table is now added to its caption, including the missing symbols that were mentioned.

  8. p18, below Fig. 11: The conclusion on correlated systematics is that they have a negligible impact. Yet it is later stated that they cannot be ignored, which is a bit contradictory. Perhaps rephrase this or remove this statement, since it seems like they can be ignored in this case. → Their impact is negligible if theory uncertainties are included since those effects are too large. Once the theory unc. are decreased, they are important to consider. With this differentiation, correlations can be neglected under certain circumstances but not in general.

  9. p19, before Sec. 4.3: Can you say anything about the impact of neglecting correlations between theory uncertainties? Or is this not possible due to the way that you implement them in SFitter? I believe that theory uncertainty correlations are often neglected in other works, so it would be interesting to see their impact, given the fact that theory errors dominate in this study. → Unfortunately we are not able to fully correlate theory uncertainties in SFitter. Thus far we only correlate the different systematic uncertainties. But, we agree, it would be interesting to see the effect of correlated theory uncertainties. This could be part of future studies.

For a better display of the changes regarding your comments and suggestions, we marked them in green in the attached version of the paper.

Attachment:

diff_bTAFSoT.pdf

Author:  Nina Elmer  on 2025-01-09  [id 5099]

(in reply to Report 2 on 2024-07-19)
Category:
answer to question
correction

We thank the referee for the careful revision of our manuscript. Here, we address the various points raised in their reports and we explain how the manuscript has been modified.

In many places, I found the language somewhat vague and colloquial, bordering jargon. A revised version would profit from a leaner and more crisp language. → We had a careful look at the language and changed it whenever we identified jargon.

I found the description of the construction of the likelihood (p9) too vague. → We changed the description whenever we thought additional information was helpful and added a comment on flat likelihoods.

More detailed comments 1) Table 2: There are several potentially important measurements not included. ((very new) total+differential tt/tt+jets in lepton+jets ATLAS https://arxiv.org/abs/2406.19701, total+differential ttW ATLAS https://arxiv.org/abs/2401.05299, total+differential ttW CMS https://arxiv.org/abs/2208.06485, tt+gamma 13 TeV ATLAS https://arxiv.org/abs/1812.01697 https://arxiv.org/abs/2403.09452, tt+gamma 13 TeV CMS https://arxiv.org/abs/2107.01508 https://arxiv.org/abs/2201.07301 The ttbar spin correlation measurement provides tt(2l) unfolded cross sections that could constrain ctG (your Fig 10) better than the combination. → We agree that some of these analyses might be useful, but especially the differential tt+jets measurement was published long after the original version of the paper. Total rates and/or low-rate associated channels are unlikely to affect our analysis much. We decided to stick to the analyses included in the original paper and will consider these analyses in our next, dedicated SFitter top sector analysis.

2) p7/8, Fig. 1. I found the provided examples of the SMEFT sensitivity relatively short. It is not possible to show all possible variations, but a few more examples, possibly with other operators would be very interesting. Why not put all the inclusive cross-sections in a figure, one in each bin, and show how these change with SMEFT by overlaying a few selected parameter points? → We are sorry, but we are not sure what the referee is referring to. Fig.1. illustrates the effect of one especially interesting operator on a kinematic distribution. That is really the only point we feel we need to discuss further, more information was included in our comprehensive analysis 1910.03606.

3) You emphasize correctly in the Appendix that the Monte-Carlo replica method has shortcomings, related, in particular, to the quadratic SMEFT terms. But in Eq. 12 you use the asymptotic Gaussian assumption for the Markov Chain. This seems to imply Wilk/Wald which could be violated by quadratic terms as discussed in arxiv:2207.01350. The Gaussian approximation could be invalid in this case. For example, Fig. 1 shows positive modifications for positive and negative values for CQd8, indicating that quadratics are dominant at the 3sigma level. It should be checked (or explained) why/how/if the quadratic terms invalidate the statistical theorems. (I believe the tests in, e.g., Sec. 3.1 are linear in the POI and would are not conclusive in this regard.) → We use the Gaussian approximation for the statistical part of the SFitter likelihood for the top sector, because it simplifies things, and because we convinced ourselves that, unlike in the Higgs sector, we do not have to use the general Poisson form. This affects only the construction of the fully exclusive likelihood, the marginalization and the profiling are discussed in the Appendix. We have not found any fundamental issues with the validity of these two approaches, once we keep in mind that they can give different results, for instance in the presence of flat theory uncertainties. This means we are not confident to comment on fundamental violations of statistical theorems through, for us, just numerical simplifications. We don’t use toys in our SFitter implementation

4) I think Section 2.3 can be improved in terms of the quality of the text and the level of detail in the construction of the likelihood. In particular, p9 and the paragraph starting with "By ansatz..." can be significantly improved. It is not quite clear how theory uncertainties are correlated. Even though the treatment is heuristic, they should be correlated across bins/measurements for each process separately. There is too little detail provided on how the theory uncertainties are obtained (which generator scale choice?). → We went through the text again carefully and adjusted the text where necessary. We provide some additional information on the construction of the likelihood and added a sentence on theory uncertainties and their correlation across different bins and measurements. Furthermore, additional details on how scale uncertainties are obtained have been included.

5) (Section 3) I have no substantial concerns with the studies of the public likelihood. However, the nuisance parameters appearing in the 2D distributions and the naming of the uncertainties in the impact plots should be harmonized. The uncertainties should also be explained briefly in the text. → Only the names for the parameters in the impact plots are provided and are used to allow for an easy comparison. The 2D distributions show nuisance parameters where these names are not provided by experiment so we cannot harmonize these properly. We included a brief comment on this in the text.

6) (Section 4.4) This section should be improved and better integrated in the paper. Several operators appear only in the appendix. It emphasizes OtG, but the most sensitive measurements (see above) are not included. It appears the Higgs combination is a late addition to the draft and stands rather alone. As for the fit in the top sector, lists of operators, uncertainties, etc. should be added. → We went through the section again carefully and adapted the text when necessary. A detailed list of operators for the Higgs and diboson sector can be found in Appendix B or in 2208.08454, as mentioned in the paper.

Minor comments: p2, 2nd paragraph: SMEFT is renormalizable, but not in the sense of the SM which requires a fixed and unchanged number of counter terms to all orders. Please clarify. → That is a good point, but since we are truncating terms beyond dimension six, it makes SMEFT renormalizable again, because we have a finite number of operators and counterterms. This truncation is motivated by the fact that contributions beyond dimension six are negligibly small in a global analysis. As we also state in the second paragraph on page 3.

p2, 3rd paragraph. Many more references than [45] (HighTEA, with 3 references itself) would be needed or the sentence should be changed so that it actually refers to what is done in [45]. → When consulting the High-Tea website, they suggest to only cite this one paper when using it. As mentioned in the text, HighTEA is only used for the K-factor calculation of the ttbar production.

p3, Eq. 1: It looks odd to use the same index k (without bounds) to count two different sets of operators. Moreover, the Weinberg operator (dim5) is not mentioned at all - it should be said that it is not relevant. → We now explicitly mention that the Weinberg operator is neglected and make the notation of the sums more clear.

"Because the underlying symmetry structure is an input to the EFT construction" ... Do you want to say that SMEFT operators have well-defined CP properties and therefore you can remove the CP violating sector? Is the "symmetry structure" the discrete SM symmetries? This paragraph should be made clearer in this regard. → Indeed, the EFT expansion starts with an assumed set of fields and symmetries and then includes all allowed operators at a given dimensionality. We clarified this in the text.

Table 1, Eq.2 What about 2-quark-two-lepton operators? Those affect tt+multilepton final states and are not mentioned. → Sorry for being confused. In the top pair signal such operators would be strongly suppressed off-shell contributions. In the top decays additional operators can be included, but the top decays themselves are limited in the momentum flow and therefore not especially sensitive. We clarified this in the text.

Eq. 6 \mp -> \pm should be more appropriate → We changed it to \pm.

Sec. 2.3 3rd paragraph: I do not understand the 2nd sentence. You do truncate the Lagrangian but also the LHC rate predictions at the quadratic level. → We do indeed truncate both the Lagrangian and rate predictions at the quadratic level. We emphasize that the truncation of the Lagrangian removes any dimension 8 effects from these quadratic contributions.

p9 3rd paragraph. "allowed shift ... at no cost in the likelihood" is unclear. I think you refer to an (additive or multiplicative?) nuisance parameter whose impact is inexpensive to compute. → We rephrased the statement in question to make it more clear.

Sec. 4.1 There are several unnecessary forward references in the text, e.g., in the first sentence. The last two paragraphs (in particular the last sentence) in the outlook do not integrate well with the narrative of the rest of the text. → We went carefully through the text again, and removed any forward references we considered unnecessary.

For a better display of the changes regarding your comments and suggestions, we marked them in orange in the attached version of the paper.

Attachment:

diff.pdf

---

## Round 3 · Referee Report · Anonymous (Referee 1) · 2025-2-12

Report

I want to thank the authors for furnishing this update. The clarity has improved substantially. I have no further concerns and recommend the manuscript for publication.

Recommendation

Publish (surpasses expectations and criteria for this Journal; among top 10%)

---

## Round 3 · Referee Report · Anonymous (Referee 2) · 2025-2-19

Report

I am satisfied with how the authors have addressed my suggestions and questions. No further modifications are required on my side.

Recommendation

Publish (surpasses expectations and criteria for this Journal; among top 10%)

---

## Round 3 · List of Changes

We would like to thank the referees for their comments and provide an updated version addressing their requests. A detailed list of changes can be found in our replies to the referees.

---

## Editorial Decision

published